# Ordinary kriging as a tool to estimate historical daily streamflow records

William H. Farmer[1]

[1]U.S. Geological Survey, Box 25046, Denver Federal Center, MS 410, Denver, CO, 80225, USA

*Correspondence to:* William Farmer (wfarmer@usgs.gov)

**Abstract.** Efficient and responsible management of water resources relies on accurate streamflow records. However, many watersheds are ungaged, limiting the ability to assess and understand local hydrology. Several tools have been developed to alleviate this data scarcity, but few provide continuous daily streamflow records at individual streamgages within an entire region. Building on the history of hydrologic mapping, ordinary kriging was extended to predict daily streamflow time series on a regional basis. Pooling parameters to estimate a single, time-invariant characterization of spatial semivariance structure is shown to produce accurate reproduction of streamflow. This approach is contrasted with a time-varying series of variograms, representing the temporal evolution and behavior of the spatial semivariance structure. Furthermore, the ordinary kriging approach is shown to produce more accurate time series than more-common, single-index hydrologic transfers. A comparison between topological kriging and ordinary kriging is less definitive, showing the ordinary kriging approach to be significantly inferior in terms of Nash-Sutcliffe model efficiencies while maintaining significantly superior performance measured by root mean squared errors. Given the similarity of performance and the computational efficiency of ordinary kriging, it is concluded that ordinary kriging is useful for first-order approximation of daily streamflow time series in ungaged watersheds.

## 1 Introduction

One of the most fundamental problems confronting the fields of hydrology and water resources management is the prediction of hydrologic responses in ungaged basins (PUB) (Sivapalan et al., 2003). While streamgages have long provided point measurements of the daily time series of streamflow, there are many regions of the globe that remain sparsely gaged, and thus, there are many completely ungaged locations (for an example in the US, see Kiang et al., 2013). Building on the long history of hand-drawn maps showing the spatial variation of hydrologic and climatic variables, geostatistical techniques are proposed as a means of leveraging the information content of streamgage networks to produce spatially and temporally continuous predictions of historical daily streamflow. The primary goal of this work is to demonstrate that simple geostatistical techniques can provide predictions of daily streamflow time series at ungaged sites that are superior to those produced by the single-index, transfer-based techniques. It is also hypothesized that simple geostatistical techniques produce estimates nearly as good as those produced by more advanced geostatistical tools.

Techniques for the reproduction of historical records of streamflow largely fall into two main categories: process-based models and transfer-based, statistical techniques. This work is concerned with the latter, which rely on transferring information

from an index site or set of index sites to an ungaged site by the means of a statistical relationship. These techniques include ungaged applications of record reconstruction techniques like the drainage-area ratio method (see Asquith et al., 2006), the maintenance of variance extension (Hirsch, 1979, 1982), and nonlinear spatial interpolation using flow duration curves (Fennessey, 1994; Hughes and Smakhtin, 1996). A portion of this work is dedicated to contextualizing geostatistical techniques within these traditional approaches.

The prediction of daily streamflow records in ungaged basins, especially for statistical transfer methods, has largely been dominated by one-to-one transfers from an index streamgage to an ungaged site (as in Archfield and Vogel, 2010; Farmer et al., 2014). In some cases, information from a few neighboring streamgages has been blended to predict values at an ungaged site (Andréassian et al., 2012; Shu and Ouarda, 2012). Since not all streamgages are used to produce predictions, these approaches neglect some of the information content of the streamgage network. Alternatively, regional hydrologic methods have sought to incorporate information from all the gaged sites to produce regression equations (Vogel et al., 1999) or contour maps (Sauquet, 2006) describing the spatial variation of hydrologic variables of interest. It is hypothesized here that predictions of daily streamflow time series can be improved by incorporating regional information beyond the information available at single index streamgages and that, building on previous hydrologic time series analysis (Solow and Gorelick, 1986; Skøien and Blöschl, 2007), this can be achieved by utilizing the geostatistical method known as kriging.

Geostatistical tools have been used to develop regional maps of measured and predicted hydrologic and climatic variables for decades. The U.S. Geological Survey has developed contour or isoline maps of runoff in the United States as far back as 1894 (Langbein, 1949). Langbein (1949) provide a summary of early hydrologic mapping efforts in the United States and elsewhere dating back to 1873. Such efforts produced largely hand-drawn maps of runoff, precipitation, and evapotranspiration that relied heavily on expert judgment rather than algorithmic geostatistics (Langbein, 1949; Busby, 1963). As researchers gained access to higher-powered computers, efforts were made to automate the development of maps of mean annual runoff (Langbein and Slack, 1982). In both Europe and the United States, maps of mean annual runoff generated by geostatistical techniques were found to be as accurate as their hand-drawn predecessors (Rochelle et al., 1989; Domokos and Sass, 1990; Bishop and Church, 1992, 1995). Mapping techniques have also been used to explore other streamflow statistics (Gottschalk et al., 2006; Archfield et al., 2013) and to assess the accuracy and performance of hydrologic models (Sauquet and Leblois, 2001).

Geostatistical maps of runoff and other variables are usually based on kriging, a technique developed in the mining industry (as described by Skøien et al., 2006). In kriging, the predicted variable is considered to be spatially continuous and predictions are based only on geospatial locations. A method known as co-kriging can also be used to introduce variables beyond geospatial locations into the prediction. The use of geospatial locations is generally valid for variables like precipitation and temperature, but runoff is different. Streamflows are organized hierarchically along a stream network and typically conserve mass (Sauquet et al., 2000; Sauquet, 2006; Skøien and Blöschl, 2007). For this reason, topological kriging (top-kriging) was developed to incorporate the river network and its geographic extent into kriging estimates (Bishop et al., 1998; Sauquet et al., 2000; Sauquet, 2006; Skøien et al., 2006). In studies exploring the prediction of mean annual runoff (Skøien et al., 2006), percentiles, and other indices of the streamflow distribution (Castiglioni et al., 2011; Archfield et al., 2013) and streamflow signatures (Viglione et al.,

2013), top-kriging has been shown to outperform many other techniques, including ordinary kriging. However, ordinary kriging is better understood than top-kriging and, according to Sauquet (2006), may provide a competitive first-order approximation.

Despite its wide application for the prediction of streamflow statistics, kriging, top-kriging, and mapping in general have not widely been used to predict time series of streamflow and related variables. Despite the need for sub-monthly predictions of
streamflow statistics, the prediction of sub-monthly variables was originally thought to be computationally prohibitive (Arnell, 1995). Previous work (Solow and Gorelick, 1986) showed kriging could be used for monthly time series prediction. With advances in computer technology, Skøien and Blöschl (2007) applied top-kriging to the prediction of hourly time series of runoff in Austria. Though they did not compare their techniques to ordinary kriging, they found that the embedded network structure of top-kriging produces good estimates of the runoff time series, but that additional spatial and temporal improvements, such
as the inclusion of complex river routing and lag times, yielded diminishing returns. Aggregating their hourly model to daily estimates, they showed that top-kriging was superior to a deterministic rainfall-runoff model. Because it has not been previously considered, it is important to explore and contrast the potential of ordinary kriging and top-kriging to predict streamflow time series in ungauged basins.

This work explores the potential of ordinary kriging to produce spatially and temporally continuous predictions of historical
daily streamflow in the southeast region of the United States. Streamflow is a volumetric quantity that typically accumulates along a river network; as it is not reasonable to consider the regionalization of a volumetric quantity, a transformation is needed. This has been the rationale for the prediction of unit runoff values (Skøien and Blöschl, 2007), where unit runoff is defined as the ratio of streamflow to drainage area. Here, kriging is used to predict a time series of the same variable, as it is both spatially continuous and can be back-transformed to produce volumetric streamflow predictions.

Spatial interpolation driven by semivariance – kriging – among daily streamflows is not new. Skøien and Blöschl (2007) used a single, temporally-aggregated representation of spatial correlation to predict all daily values. Similarly Archfield and Vogel (2010), in their map correlation method, leverage the spatial correlation structure of hydrographs in streamgage networks to identify ideal index streamgages. This work presents a different approach exploring the temporal evolution of daily variograms and seeking to characterize the spatial correlation of daily streamflows in a region. This work evaluates the ability to estimate
daily streamflow series at ungaged sites. Using a leave-one-out validation procedure, the predicted time series of daily flows at gaged-but-omitted sites are assessed across a range of goodness of fit metrics. Furthermore, the temporal evolution and stationarity of the spatial semivariance structure of daily streamflow is explored through time-series analysis. It is shown that ordinary kriging of the logarithms of unit runoff can provide accurate streamflow predictions at ungaged sites, significantly outperforming more traditional approaches that employ a single index streamgage for transfer. The work presented in this
manuscript is an extension of the material presented by Farmer (2015).

## 2 Data and Methodology

### 2.1 Study Area and Streamflow Data

Using a data set identical to that used by Farmer et al. (2014), this analysis was conducted with data from 182 streamgages in the southeastern United States. Basin characteristics are summarized in Table 1 of Farmer et al. (2014), but drainage areas averaged 979 km$^2$. The range of drainage areas was from 14 to 38 849 km$^2$, with a median of 417 km$^2$ and first and third quartiles of 150 and 886 km$^2$. Because the basins are free of major regulation or development, all of the streamgages were considered near reference-quality according to their designation in the GAGES-II classification (Falcone, 2011) or their local approval and utilization in previous flood-frequency studies (Gotvald et al., 2009). The two sources provide a more thorough description of their criteria. Figure 1 shows the geographic extent of the study area and streamgage locations, which are defined by the Albers projection, in meters, of the latitude and longitude of basin outlet with respect to the North American Datum of 1983. As described by Farmer et al. (2014), the 355 000-km$^2$ study area, covering portions of seven Southern states, is warm, humid, temperate, and nearly 50% forested. Only 9% of the landscape is categorized as developed, while 18% is occupied by agricultural uses.

Daily streamflow records were obtained from the U.S. Geological Survey National Water Information System (http://waterdata.usgs.gov) for the period from 01 October 1980, through 30 September 2010. As documented by Farmer et al. (2014), very small portions of the streamflow records – for periods ranging from one to 33 days long – were reconstructed using standard techniques. To avoid the complications of zero values, zero-valued streamflows were assigned a value of 0.00003 cubic meters per second, a value smaller than the smallest streamflow reported by the U.S. Geological Survey. Farmer et al. (2014) and Farmer (2015) found that this substitution had only a minimal effect on the interpretation of results. A full description of their data set, which was used without additional modification, is presented by Farmer et al. (2014). Across the 182 streamgages considered, there were 1.6 million observations of daily streamflow. Contained at only seven of the 182 sites, 5 435 observations were zero, an occurrence of only 0.3%. If zero values were more prevalent, they may have hard a substantial impact on the results presented herein.

### 2.2 Ordinary Kriging

Ordinary kriging is a geostatistical tool by which the distance between two points is used to predict the semivariance of some dependent variable. The inter-site semivariances of data from a measured network can be used to create a system of linear equations predicting the semivariance at an unmeasured site to be a linear sum of the semivariance between all observed sites. For an unmonitored site, this allows for the derivation of linear weights between the unmonitored site and all monitored sites in the network. If all the assumptions of ordinary kriging are valid, this tool provides the best linear unbiased estimate.

Journel and Huijbregts (1978), Isaaks and Srivastava (1989), Cressie (1993), Skøien et al. (2006), Archfield and Vogel (2010) and many others provide an elegant and simple description of the mathematics of kriging; only a summary of the general principles is provided here. Consider a network of measurements $z(\boldsymbol{x}_i)$ for $i = [1,...,n]$, where $\boldsymbol{x}_i$ is the location of the measurement. Ordinary kriging allows for the prediction of an unmeasured value at location $\boldsymbol{x}_0$, $z(\boldsymbol{x}_0)$, by calculating a

weighted sum of the observations $\hat{z}(\boldsymbol{x}_0) = \sum_{i=1}^{n} \lambda_{i,0} z(\boldsymbol{x}_i)$. The kriging weights, $\lambda_{i,0}$, for a particular ungaged location are determined by solving the linear system

$$\boldsymbol{\gamma}\boldsymbol{\lambda_0} = \boldsymbol{\gamma_0} \tag{1}$$

for the vector of weights, $\boldsymbol{\lambda_0}$, where

$$\boldsymbol{\gamma} \equiv \begin{cases} \gamma_{i,j} = \dfrac{1}{2}(z(\boldsymbol{x}_i) - z(\boldsymbol{x}_j))^2 & \text{for} \quad i,j \leq n \\ \gamma_{i,n+1} = \gamma_{n+1,j} = 1 \\ \gamma_{n+1,n+1} = 0 \end{cases} \tag{2}$$

$$\boldsymbol{\lambda_0} \equiv \begin{cases} \lambda_{0,i} = \lambda_{i,0} & \text{for} \quad i \leq n \\ \lambda_{0,n+1} = \mu \end{cases} \tag{3}$$

and

$$\boldsymbol{\gamma_0} \equiv \begin{cases} \gamma_{0,i} = \frac{1}{2}(z(\boldsymbol{x}_i) - z(\boldsymbol{x}_0))^2 & \text{for} \quad i \leq n \\ \gamma_{0,n+1} = 1 \end{cases} \tag{4}$$

This system ensures that all the weights sum to one and estimates the LaGrange multiplier, $\mu$, to control for the unknown mean of $z$.

The single realization of $\gamma$ that is produced from the sample observations of $z$ cannot be considered to represent the underlying system. The sample may produce a matrix that is singular or not positive definite, conditions required for solution of the system. Furthermore, the elements of $\boldsymbol{\gamma_0}$, by nature, are unobservable as the value of the dependent variable at the un-

gauged location, $z(\boldsymbol{x}_0)$, is what is being estimated. However, with additional assumptions of stationarity, the semivariance can be modeled as a function of separation distance. Several classical models are available to ensure positive definiteness. These models are parameterized by calibration to the empirical variogram of observed semivariance as a function of distance. Once a variogram model is selected, the system becomes

$$\hat{\boldsymbol{\gamma}}\boldsymbol{\lambda_0} = \hat{\boldsymbol{\gamma_0}} \tag{5}$$

which is solvable. The resultant weights can then be used to estimate the dependent variable at the ungauged site.

Ordinary kriging of streamflow time series builds off of previous hydrologic applications to predict streamflow statistics to produce a method for handling temporal variation along with spatial variation. Based on initial exploration by the Farmer (2015), the spherical variogram model was selected for the application presented here. In formal terms, the semivariance is represented as

$$\gamma(h) = \frac{1}{2}E[(z(\boldsymbol{x}+h) - z(\boldsymbol{x}))^2] \tag{6}$$

where $x$ is a geospatial location and $h$ is a separation distance. The spherical variogram model approximates the semivariance as

$$\hat{\gamma}(h) = \begin{cases} (\sigma^2 - \tau^2)(\frac{3h}{2\phi} - \frac{h^3}{2\phi^3}) + \tau^2 & \text{if} \quad h \leq \phi \\ \sigma^2 & \text{if} \quad h > \phi \end{cases} \qquad (7)$$

where $\sigma^2$ is the partial sill, $\phi$ is the range and $\tau^2$ is the nugget variance. Alternative models are available, but Farmer (2015) and initial testing done here found the results to be generally insensitive to the variogram model type. The spherical model has been used previously for hydrologic phenomenon (Archfield and Vogel, 2010). Here, this model was developed with dependent variable as the logarithm of the measured streamflow per unit drainage area, $z = \ln \frac{Q}{A}$. Previous work (Farmer, 2015) found that this dependent variable was the most stable predictand. Even though this logarithmic transformation was used, several performance metrics were assessed by considering exponentiation as the simple back transformation without an attempt at bias correction. Finally, in building the empirical variogram, the semivariances were stratified into ten equal-interval groups based on the inter-site distances ranging from zero to the maximum inter-site distance of 920 km, as suggested by Archfield and Vogel (2010). The solution of this kriging system was implemented using the geoR package (Ribeiro and Diggle, 2015).

The model of ordinary kriging presented above assumes a global neighborhood. That is, all observations are assigned a weight for the prediction of the ungaged site. In other hydrologic applications (Pugliese et al., 2014), some advantages have been gained by restricting the number of sites permitted to influence predictions. The neighborhood can be restricted to include only the $k$ nearest neighbors. This approach was considered, but results were found to be generally insensitive to the number of neighbors. As a result, the global neighborhood was used, allowing the kriging algorithm to minimize the weights of far-distant sites if they are unimportant for estimation.

While there are many considerations in the development of a kriging system, this work is mainly focused on kriging time series and the temporal behavior of kriging parameters. As such, the temporal evolution and behavior of variogram parameters was of most interest. As discussed above, there are many considerations in the development of a kriging system. Several were explored, including the binning of empirical variograms, the number of contributing neighbors, and the maximum range of the variogram, but none were found to have only a marginal impact on the resulting estimates. Accordingly, the remainder of this manuscript considers the unique problems of temporal calibration and prediction.

## 2.3 Variogram Parameters

The variogram can be characterized by three parameters: The nugget value, partial sill, and the range. The nugget value is the semivariance of collocated points or, as it is sometimes interpreted, the measurement error, the partial sill represents the regional semivariance, and the range represents the separation distance beyond which the inter-site semivariance is best approximated by the regional semivariance. In some previous hydrologic applications of kriging, the semivariance, which is modeled by the semivariogram, has been assumed to be temporally constant and thus only a single variogram model need be fit. This is clearly not the case for the reconstruction of historical time series of streamflow. It is therefore important to consider the

temporal evolution, or lack thereof, in the spatial semivariance structure, as characterized by variogram parameters, of daily streamflows.

The initial development by Farmer (2015) modeled each day of the streamflow record independently with unique variogram parameters. While this proved useful, it is not intuitive because a basic understanding of hydrology suggests a strong temporal dependence across daily streamflows. With the temporal dependence of streamflows, it seems reasonable to consider some corresponding temporal dependence in variogram parameters. As an end-member along the continuum of parameter smoothing, Farmer (2015) showed that assuming temporal stationarity in variogram parameters resulted in barely any degradation of performance: The average of the daily variogram parameters, which are not identical to the pooled variogram parameters (described below), performed nearly as well as the independent daily models.

This work considers the temporal evolution of variogram parameters more formally. The streamflow models based on independent daily variogram models are contrasted with a pooled variogram model. The latter model requires the fitting of only a single variogram, while the former requires the fitting of as many variograms as there are days to be simulated. If the parameters of the semivariogram can be reasonably assumed to be constant, then the computational efficiency of the pooled model is highly advantageous for operational prediction.

For a daily variogram, the semivarainces for each site pair are plotted against distance, binned, and averaged to fit a variogram model; the process is repeated independently for each day. The pooled variogram is described by Gräler et al. (2011). For pooled variograms, the semivariances calculated on each day are pooled into a single empirical variogram to which the variogram model is calibrated. The semivariance is calculated spatially, as described above, but semivariances between sites are not computed across time steps. That is, $cov(z(\boldsymbol{x}_{i,t_1}), z(\boldsymbol{x}_{j,t_1}))$ and $cov(z(\boldsymbol{x}_{i,t_2}), z(\boldsymbol{x}_{j,t_2}))$ are both considered and pooled into the empirical variogram cloud, but $cov(z(\boldsymbol{x}_{i,t_1}), z(\boldsymbol{x}_{j,t_2}))$, where the time $t_1 \neq t_2$, is never considered. In section 2.4 and elsewhere, Gräler et al. (2011) describe and contrast the performance of the pooled method and the averaging method. The average model treats each empirical variogram equally, while the pooled model weights each bin by the number of pairs in each bin of the variogram cloud. The similarities identified by Gräler et al. (2011) suggest that the averaged model considered by Farmer (2015) can be represented much more efficiently by the pooled model. However, averaging variogram parameters will not necessarily lead to the same model as fitting a model to averaged or pooled empirical variograms If all streamgages are operational on all days, then the average model is identical to the pooled model.

## 2.4  Relative Performance

In addition to contrasting temporally independent variograms and pooled variograms, this paper also contrasts these methods with two standard, transfer-based statistical tools: the drainage area ratio (DAR)(Asquith et al., 2006) method and nonlinear spatial interpolation using flow duration curves (QPPQ) (Hughes and Smakhtin, 1996). The former scales index streamflows by drainage areas, while the latter scales the entire flow duration curve of an index site. Both of these methods were implemented following the methodology of Farmer et al. (2014). The time series prediction methods were assessed by computing the Nash-Sutcliffe model efficiency (Nash and Sutcliffe, 1970) of the streamflow values and the logarithms of streamflow values. Nash-Sutcliffe model efficiencies range from one to negative infinity; values of one indicate a perfect model fit, while lower

values indicate an increasingly poor fit; a value of zero indicates that the estimate is no better than a regional average. Pearson correlations between observed and simulated streamflows, root mean squared errors and average biases were also considered.

Previous work (Pebesma et al., 2005; Gupta et al., 2009; Gupta and Kling, 2011) has shown the dependencies between Pearson correlation, root mean squared errors and the Nash-Sutcliffe model efficiency. Though inter-related, all metrics are
included here to highlight the components of the model efficiency and to more deeply appreciate the strengths and weaknesses of each method. Additionally, Gupta et al. (2009) showed that the skewed distribution of daily streamflow may substantially alter the interpretation of the Nash-Sutcliffe model efficiency. For this reason, it is important to understand how the component parts of the Nash-Sutcliffe model efficiency, namely the Pearson correlations and root mean squared error, vary independently. Even observing any disagreements across metrics, the Nash-Sutcliffe model efficiency, by removing some of the skewness of
daily streamflows, may provide a more reliably interpretable metric.

As is described below, the kriging methods were implemented to predict a logarithmic transformation of streamflow. With the exception of the Nash-Sutcliffe model efficiency of the logarithms themselves, all other performance metrics were computed on back-transformed streamflows. No bias correction factor was developed or applied. The development of a bias correction factor that can be applied to ungauged basins is beyond the scope of this work but is essential to future explorations.
Using the same metrics, ordinary kriging was contrasted with an application of top-kriging similar to that defined by Skøien and Blöschl (2007). Top-kriging was applied using the rtop package (Skøien, 2015), which uses spatial regularization rather than the spatio-temporal regularization presented by Skøien and Blöschl (2007). The differences can be assumed to be negligible for this application. Regardless, here, top-kriging was applied with a minimum spatial resolution of 100 points per basin and a maximum of 5 neighboring basins per prediction. Furthermore, the daily semivariances were pooled to create a pooled
top-kriging model of the spatial semivariance structure. This comparison of ordinary and top-kriging serves as only an initial comparison. It does not address deeper levels of discrepancy between the two methods, a topic which, given the similarity of results, may warrant further research. This comparison also does not explicitly address questions of computational efficiency, a difference in which may favor one method over another.

The implementation of top-kriging presented here is not intended to represent the ultimate implementation of top-kriging
for this region. Ordinary kriging is an extreme of top-kriging in that top-kriging allows for a variable spatial support for each observation while ordinary kriging provides only one regularization point for each observation. With this in mind, this implementation of top-kriging is meant to reflect the improvements achieved by allowing for a further discretized spatial support. Certainly, the improvements of either method may be improved by considering a more robust exploration of the underlying variogram model, the number of contributing neighbors or the level of spatial discretization. However, this was left
for future research, allowing this work to focus only the effects of additional spatial discretization.

# 3 Results & Discussion

## 3.1 Optimal Variogram Parameters

In a leave-one-out validation procedure, both the daily and pooled parameter sets reproduce historical daily streamflow records quite well. Table 1 summarizes several common performance metrics calculated on the complete water years of observed daily streamflow. (A water year is the 12-month period 01 October through 30 September designated by the calendar year in which it ends.) For all metrics, the performances are very similar, but the pooled parameter set produced slightly better results. A two-sided Wilcoxon signed-rank test for each performance metric showed this difference to be significant in all cases except median bias. Figure 2 shows a one-year example of the predicted and observed streamflows for a single site; this site and year were selected because the results are representative of median performance. This example highlights the similarity between estimates made with the daily and pooled variograms, but also demonstrates the poor performance during low-flow periods. This is interesting, as some recessions are reproduced well (January through March), while others (May through June) are reproduced poorly. General biases will be discussed below, but further research is needed to more accurately understand bias in particular streamflow regimes.

In addition to having similar point performance metrics, the daily and pooled variograms produced nearly identical distributions of absolute percent errors (Fig. 3). This summary plot shows the cross-site median cumulative distribution of absolute percent errors. Both daily and pooled variograms perform well, with more than half of the estimates within 30% of the observed streamflows. Though the differences between the curves from the pooled and daily variograms are not significant, the pooled variogram produces estimates with slightly fewer large percent errors.

Figure 4, binning with a width of one percentage point, plots the cross-site median percent error against observed non-exceedance probability. The result shows a concerning limitation of the kriging approach. From Table 1, both sets of estimates produced only a slight upward bias overall – 4.5% for the daily variograms and 2.5% for the pooled variogram – but the overall statistics do not capture the poor performance in the tails of the streamflow distribution (Fig. 4). Estimates appear to be nearly unbiased, plus or minus 5%, for streamflows that are not exceeded between 5 and 76 percent of the time. For low streamflows, those not exceeded less than 5% of the time, both variogram methods consistently overestimate streamflows with percent errors between 5% and 15%. For high streamflows, those not exceeded more than 76% of the time, streamflows are actually underestimated; the underestimate approaches -40% for some of the greatest streamflows. These substantial biases in the extremes are a symptom of modeling smoothing that results from attempting to approach unbiased central tendencies when compared with observations.

Finally, with the use of time-varying and time-invariant variograms, it is useful to consider how well the temporal structure of the daily streamflows is reproduced. Figure 5 summarizes the median observed autocorrelation of streamflows and how well it is reproduced. Again, both variogram methods produce similar results, both slightly overestimating the magnitude of autocorrelation. The differences, however, are small. Because of the dependent structure of daily time series, it is not surprising that simulated results would produce some aberrant residual correlation at long time lags. In general, the reproduction of the autocorrelation structure suggests that the temporal structure of the streamflow time series is reproduced tolerably well. Not

surprisingly, the daily parameter set, which varies in time, more accurately reproduces the temporal structure. Interestingly, the difference is not as large as might be expected.

## 3.2 Temporal Evolution of Variogram Parameters

Because the pooled variogram parameters produce results fairly similar to the daily parameter sets, it is important to understand how the pooled parameters relate to their daily counterparts and how the daily counterparts evolved over time. Figure 6, described below, illustrates the temporal structure and seasonal nature of the daily parameters and contextualizes the pooled parameter sets. For each parameter, its 31-day moving median is presented in lieu of the widely variable daily values. The moving-median values are presented because the daily values exhibit dramatic extremes and fluctuations, making graphical display unintelligible. The temporal variability in variogram parameters is a reflection of the temporal and regional variability in streamflow and the factors producing streamflow.

As mentioned previously, the nugget value can be thought of as the semivariance of nearly co-located points. In the context of basins and daily parameters, the nugget on each day, because the semivariance of co-located points is akin to a variance, is an approximation of the average of all at-site variances for that day. The 31-day moving median of the nugget time series suggests that there is a substantial seasonal trend. The nugget, or regional variability, and the variability thereof, are fairly constant from the beginning of January through May and rise to a peak in September and October. The pooled parameter, which can be thought of as a time-averaged variability of an average site, is closer to the peak of the moving-median nugget than to the lower stable January-May values. The pooled parameter is greater than the median of the daily values. This suggests that, for much of the year, the pooled nugget, being greater than the daily values, introduces more daily variability than would be expected. As measurement uncertainty may fluctuate, the fluctuations in the nugget may be tied to fluctuations in the magnitude of streamflow.

The partial sill, a limit on the regional semivariance, shows a much weaker seasonal signal. The 31-day moving median shows a nearly binary structure of two values. The partial sill is small from January through March, transitions quickly in April, remains high through October and then returns towards January values. Again, the pooled parameter plots closer to the higher plateau of the moving median. This means that for parts of the year, the pooled parameter assumes the more distant neighbors hold appreciably less information than they really contain. For a smaller portion of the year, the pooled parameter, being greater than the daily values, assumes the more-distant neighbors hold slightly more information than they really do. However, the pooled partial sill remains within the inter-decile range of the daily parameter values for the majority of year. As with the nugget, fluctuations in the sill may be tied to fluctuations in the magnitude of streamflow.

The range parameter shows the least complex temporal structure. The 31-day moving median shows that the range varies over an order of magnitude, and year-to-year variability, as shown by the inter-decile range, is consistently large. The year-to-year variability is more pronounced than the season trends. Overall, there is a slight depression in the summer months, which indicates decreased regional homogeneity and more heterogeneity in that the regional semivariance (partial sill) is reached at shorter distances. The pooled parameter is quite similar in magnitude to the median daily value and is almost completely contained by the daily inter-decile ranges.

It is difficult to consider the effects on any one parameter in isolation. The final row of Figure 6 shows the temporal variability in the ratio of nugget to sill. January through April, the nugget accounts for 20-30% of the sill, dipping to only 5% in mid-May and plateauing at about 15% of the sill through the rest of the year. The averaged parameters places the nugget at only 10% of the sill, while the pooled-parameter more faithfully represents the 15% value seen for the latter half of the year. This ratio may be closely tied to measurement uncertainty. Low streamflows, which often occur in the Winter months, are generally more difficult to measure and may result in the nugget value accounting for more of the regional semivariance. The dip in the proportion of the sill accounted for by the nugget in May may result from higher streamflows. Similarly the proportion of 15% my be emblematic of average measurement uncertainty. Further research is needed to explore this conjecture.

It is clear that there is substantial temporal structure and seasonal variation in the spatial semivariance structure of daily streamflows. Given the strong temporal dependence and seasonality of daily streamflows, this is not surprising. As with streamflow, it is extremely difficult to identify causal factors resulting in these patterns. Though not explicitly explored here, it is probable that the temporal structure is driven by climatic processes. The greater nugget value in the latter half of the year indicates increased streamflow variability, year to year, during the late Fall and early Winter. The partial sill and range interact strongly with each other, one being the threshold and the other being a sort of "time to threshold". The decreased Summer range suggests that climatic response is more homogeneous in Summer months, while the Winter and Spring rises are emblematic of increased regional heterogeneity (i.e. more localized climatic drivers of streamflow.) The partial sill demonstrates an increase regional variability, beyond the range, from late Spring through Fall; otherwise the sill is smaller, suggesting that, even beyond the range, variability is lower across the region in Winter months.

### 3.3 Relative Performance

In presenting a new model for daily streamflow reconstructions, it is useful to contextualize performance by comparing against previous methods. To this end, two common statistical, transfer-based tools for the prediction of daily time series are considered: the drainage-area-ratio (DAR) (Asquith et al., 2006) method and nonlinear spatial interpolation using flow duration curves (QPPQ) (Hughes and Smakhtin, 1996). Both are applied in a leave-one-out cross-validation with index sites defined by spatial proximity. The methods and regional regressions used here are identical to those reported by Farmer et al. (2014), though a leave-one-out validation scheme is used here. DAR is a single-index analog to the kriging approach, while QPPQ represents the optimal method for this region (Farmer et al., 2014).

The performance metrics of both DAR and QPPQ are outlined in Table 1. As concluded by Farmer et al. (2014), the QPPQ methodology performed better than the DAR technique. In this analysis, both DAR and QPPQ performed inferior to the kriging approaches. As determined by individual Wilcoxon signed-rank tests of each performance metric for estimates from each method against the estimates from pooled variograms, the pooled variograms produce results with significantly better predictive power than both DAR and QPPQ individually for all performance metrics except median bias. The estimates from QPPQ were not shown to be significantly more biased than the estimates from the pooled variograms, on average. Note that the Wilcoxon test on biases was conducted on absolute values, indicating the significance of either method being closer to the optimal level of zero bias regardless of the sign of the bias.

The comparison of the pooled ordinary kriging approach and the pooled top-kriging approach does not provide as definitive of a conclusion. The top-kriging approach provides a significantly greater Nash-Sutcliffe efficiency at the 5% significance level. However, the ordinary kriging approach yielded significantly smaller root mean squared errors. In terms of bias, top-kriging provides a significantly smaller absolute bias, but the median signed bias is slightly larger; the average bias is greater, but the average deviation from unbiasedness is smaller. There is no significant difference between ordinary and top-kriging with respect to the correlations between observed and simulated streamflows and the Nash-Sutcliffe efficiencies of the logarithms of streamflow. The disagreement of the significance of the difference in correlations between observed and simulated streamflows and the difference between Nash-Sutcliffe efficiencies is the result of the interplay of the components of the Nash-Sutcliffe model efficiency, as discussed by Gupta et al. (2009).

Based on the varied performance metrics, there is not a significant difference between the ordinary and top-kriging approaches. Aside from average performance, the quantiles of the distributions of performance appear improved for top-kriging. For example, 90% of the ordinary kriging results show a Nash-Sutcliffe model efficiency of the logarithms below 0.91, while 90% of top-kriging results are below 0.93. It is not immediately apparent why the top-kriging approach might disproportionately accept the extremes of the distribution of performance. However, the pairwise comparison of the Wilcoxon signed-rank test indicates that there is not significant evidence to reject the hypothesis that pooled ordinary kriging and pooled top-kriging produce different performances. If they are not significantly different, the additional discretization of top-kriging does not appear to produce significantly improved performance to warrant the increased complexity. Future research might also consider if the prediction variances from either method are superior; though not explored here, a more accurate prediction uncertainty may improve the usefulness of simulated streamflows.

Top-kriging was explicitly developed to address both the hierarchical nature of streamflow and streamflows aggregate dependency on contributing drainage areas (Skøien et al., 2006). Ordinary kriging ignores this structure and approaches the question of prediction as if confronted with a uniformly dependent spatial field. As mentioned earlier, this implementation of top-kriging differs from the implementation of ordinary kriging only in that top-kriging allows for the varying support of contributing drainage areas. Given the results presented here, this improvement produces nearly indistinguishable results. This is likely because the ordinary kriging approach standardized streamflows by drainage area and then computed the logarithms thereof. As evidenced by a Pearson correlation of only 0.05 between the logarithms of unit runoff and the logarithms of drainage area, standardization removed much of the dependency of streamflow on drainage areas. Removing this dependency may have dampened the improvements in performance that might have been expected from top-kriging. In a region that exhibits stronger residual dependence or a higher frequency of nested basins, the advantages of top-kriging might be more marked.

## 3.4   General Discussion

The results of this analysis demonstrate that the computationally efficient routine of pooled variogram estimation can be used to fit an ordinary kriging system that produces plausible estimates of daily time series at ungaged sites. The pooled parameter estimation, which ignores temporal variation of the spatial semivariance structure, was able to reproduce observed hydrographs more accurately than other non-kriging methods considered. Both daily and pooled kriging approaches outperformed single-

index transfers. It is intriguing that accounting for temporal variation in the variograms resulted in relatively minor changes in the kriging estimates and the performance thereof. Additionally, it is somewhat concerning that the kriging techniques show a general inaccuracy in the tails of the distribution of streamflow. The comparison of ordinary and top-kriging was inconclusive, with some metrics favoring top-kriging while others favored ordinary kriging and still others were not significantly different.

It was clearly shown that the variogram parameters, characteristic of the spatial semivariance structure, exhibit seasonal and other temporal patterns. However, the averaging that occurs when pooling daily semivariance information actually resulted in a marginal improvement in the accuracy (as measured by several metrics) of resultant streamflow time series. In initial work (Farmer, 2015), it was shown that pure averaging of variogram parameters, rather than pooling, produced estimates similarly competitive with estimates from daily variograms. It is counterintuitive that ignoring temporal variation in spatial semivariance structure would not appreciably degrade performance. Still, ignoring the temporal variation of variogram parameters produced some small degradation in the autocorrelation structure of estimated streamflows at long time lags.

Although ignoring the temporal variation in the variogram parameters did not appreciably degrade performance, it may be possible to gain some improvements while retaining computational efficiency by preserving some remnants of the observed temporal variability in variogram parameters. One option might be to consider a moving-window average of daily parameters, optimizing the advantages of temporally variable parameters, while seeking to smooth out chaotic daily behavior. Another clear avenue for future research is to evaluate the possibility of constructing a temporal model of variogram parameters. One could easily imagine monthly parameter sets or parameter sets reproduced by an autoregressive integrated moving average (ARIMA) (Box and Jenkins, 1970) model. Previous work has found only marginal advantage to incorporating complex temporal structures like streamflow travel times into hydrologic geostatistics (Skøien and Blöschl, 2007), but the temporal evolution of spatial semivariance structure was not explicitly considered. As this manuscript serves as a general introduction of ordinary kriging to time-series prediction, this work was not explored further here. In particular, temporal modeling might become increasingly advantageous when considering the problem of filling-in temporally sparse records rather than simulating completely ungauged streamflows. In such a case, it may be the temporal observations along stream contain more information that neighboring contemporary measurements.

However, contextualizing ordinary kriging in the context of other hydrologic applications of geostatistics, a brief comparison of ordinary and top-kriging was presented here. Skøien et al. (2006) introduced topological kriging to the hydrologic sciences and Skøien and Blöschl (2007) applied it to streamflow time series. Following the methods of Skøien and Blöschl (2007), a pooled top-kriging model of daily streamflows was developed and compared with the ordinary kriging approach. The comparison of ordinary and top-kriging does not provide strong evidence to favor one approach over the other. Subsequent analyses may elucidate further strengths and weaknesses, but it is not possible to dismiss either method based on the evidence presented here. The pooled top-kriging model was developed using the package provided by Skøien (2015). The need to spatially discretize the network at each time step substantially increased the computation time compared with ordinary kriging (depending on processor speeds, top-kriging required just less than three days of computation time for each site predicted, while ordinary kriging required only hours of computation time per site predicted). At the time of application, the package by Skøien (2015) did not contain a method to estimate pooled variograms directly. More recent versions do contain this functionality. Once the

computation time is brought into parity with ordinary kriging, the marginal improvements of top-kriging may be more worth-while. However, it may be that the relatively simpler formulation of ordinary kriging provides the majority of the added-value of hydrologic geostatistics.

The pooling of semivariance to produce a single set of variogram parameters implicitly assumes that the spatial semivariance
structure is constant in time. While a seasonal fluctuation may be present, that same fluctuation may occur every year with no systematic change. For the study period, water years 1981 through 2010, the time series of daily variogram parameters were indeed stationary. Following the procedures of Hirsch et al. (1982) a block bootstrapping procedure with a fixed block width of one year (365 days) and 1 000 replicates of thirty-year time series was applied to approximate the probability distribution of a seasonal Mann Kendall trend test. For all three parameters, the null hypothesis of stationarity could not be rejected. The nugget
had an approximate two-side-alternative p-value of 0.286; the partial sill, 0.184; the range, 0.178. While stationarity appears valid in this instance, it does raise an interesting question in the face of changing hydrology. Will changes in human populations, land uses, and climate significantly affect the spatial semivariance structure of daily streamflows? The daily parameter sets may be an appropriate means of testing for changing hydrology and identifying dominant processes in a region.

Pooled variogram estimation and ordinary kriging allow for the efficient and, according to broad metrics, accurate prediction
of daily streamflow at ungaged sites. Being able to regionally characterize networks of streamflow may provide additional advantages. Though not explored here, kriging algorithms also allow for the quantification of variances around estimates. This can serve two purposes: (1) It shows where in the network uncertainties are likely to be greatest, which might be a means to identify optimal locations for additional monitoring. (2) It may be able to explicitly provide confidence intervals for estimated daily streamflows. Future studies will explore the accuracy of so-derived intervals. In any case, the theoretically-
derived structure of the kriging system promises a more "closed-form" interpretation of predictive uncertainty than more traditional single-index hydrologic transfers, which require an ad-hoc procedure for uncertainty quantification. While predictive performance was indistinguishable here, more advanced methods like top-kriging may provide significant advantages in their quantification of predictive uncertainty.

One limitation of the kriging approach, as documented here, is the overestimation of the lower tail of the streamflow dis-
tribution and the underestimation of the upper tail. Similar results were documented by Skøien and Blöschl (2007). This is effectively a compression of the distribution of streamflows, resulting in estimated streamflows that are less variable than the observed streamflows. Less variability means that the estimated time series will not be able to faithfully reproduce the frequency and magnitude of the most extreme events. As the most extreme events tend to have the greatest impact on human populations, the failure to accurately reproduce them may prove problematic for operational hydrology. Interestingly, this re-
sult may not be a product of the kriging system. It may be a symptom of randomness associated with a leave-one-out validation or transformation bias, but the dramatic median suggest a more systemic problem. Instead, bias in the extremes is an expected result of deterministic modeling, whereby a single realization of simulated output is produced. If sources of error or uncertainty are neglected in order to produce such a deterministic estimate, the expectation of the conditional mean is less variable than the observed quantity. Stochastic simulation, which is possible using the predictive uncertainty of a kriging method, may be
the only solution if the estimated time series are to be made useful in the context of operational hydrology.

## 4 Summary and Conclusions

The estimation of daily streamflow records at ungaged sites is a fundamental problem of water resources management and assessment. Many tools exist to aid in quantifying resources, but this paper discusses a statistical tool that is capable of combining time series at multiple sites for regional prediction. Building on the work of hand-drawn discharge maps, ordinary kriging is proposed as an efficient technique for reproduction of historical streamflow time series at ungaged sites. Using a leave-one-out validation and daily streamflow data from 182 minimally-impacted and minimally-regulated watersheds, geostatistical techniques are shown to have advantages over other, common statistical approaches.

Ordinary kriging is demonstrated to produce more accurate streamflow time series estimates than the drainage-area ratio method and nonlinear spatial interpolations using flow duration curves. In addition, using pooled variogram parameters with ordinary kriging produced marginally better performance than using parameters determined at a daily time step. This is surprising, as pooling effectively averages out temporal variation. Though significant improvements are unlikely, it is observed that the variogram parameters, characterizing the spatial semivariance structure, show clear seasonal patterns that may be reproducible in part without requiring the computation of daily variograms. However, in an initial exploration, the advantages of moving towards a more complex kriging system such as that provided by top-kriging are, at best, minimal. Further research may improve the computational parity of top-kriging and continue to elucidate the advantages and disadvantages of ordinary and top-kriging for spatio-temporal hydrologic geostatistics.

*Acknowledgements.* This paper represents the evolution of work published as part of the author's Ph.D. dissertation. This research was supported by the Department of the Interior's WaterSMART initiative and the U.S. Geological Survey's National Water Census. Any use of trade, product, or firm names is for descriptive purposes only and does not imply endorsement by the U.S. Government. David Wolock and Gregory Koltun, both of the U.S. Geological Survey, provided valuable reviews of the initial manuscript. Edzer Pebesma, Jan Olav Skøien, and Alessio Pugliese provided valuable reviews as part of the public commentary.

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

Table 1. Summary statistics of several performance metrics for different streamflow record prediction techniques. Summary statistics include (a) the median, (b) the 10th and 90th percentiles, and (c) the Wilcoxon signed-rank probability of a difference between pooled kriging and each other method equal to or more extreme than observed (commonly referred to as a p-value). Naturally, this is not applicable (NA) for the comparison of pooled kriging against itself. Nash-Sutcliffe efficiency values of 1 indicate perfect agreement between observed and predicted values, so values closer to 1 indicate better model performance. Root mean squared error is reported in cubic meters per second. The Wilcoxon test on bias was performed on the absolute value of the bias.

| Performance Metric | | Daily Kriging | Pooled Kriging | Pooled Top-Kriging | DAR | QPPQ |
|---|---|---|---|---|---|---|
| Nash-Sutcliffe Efficiency | a | 0.7006 | 0.7040 | 0.7106 | 0.5529 | 0.5980 |
| | b | (0.3905, 0.8433) | (0.4005, 0.8522) | (0.3612, 0.8805) | (-0.2283, 0.8248) | (0.0072, 0.8026) |
| | c | 0.0068 | NA | 0.0248 | <0.0001 | <0.0001 |
| Nash-Sutcliffe Efficiency of the Logarithms | a | 0.7830 | 0.7961 | 0.7974 | 0.6663 | 0.6738 |
| | b | (0.0960, 0.9088) | (0.0935, 0.9117) | (0.2547, 0.9327) | (-0.7771, 0.8948) | (-0.2811, 0.8580) |
| | c | 0.0001 | NA | 0.0776 | <0.0001 | <0.0001 |
| Pearson Correlation Coefficient | a | 0.8562 | 0.8686 | 0.8628 | 0.8094 | 0.8148 |
| | b | (0.7004, 0.9365) | (0.7143, 0.9354) | (0.6808, 0.9542) | (0.5621, 0.9298) | (0.5842, 0.9219) |
| | c | <0.0001 | NA | 0.3330 | <0.0001 | <0.0001 |
| Root Mean Squared Error | a | 6.2733 | 5.8775 | 6.2613 | 7.2752 | 7.7550 |
| | b | (1.2685, 26.236) | (1.2775, 27.017) | (1.1633, 25.590) | (1.4270, 33.361) | (1.5690, 30.931) |
| | c | 0.0016 | NA | 0.0009 | <0.0001 | <0.0001 |
| Median Percent Bias, [Median Absolute Value] | a | 4.4727, [18.397] | 2.4686, [19.260] | 4.5863, [16.848] | 13.350, [25.252] | 8.1408, [23.191] |
| | b | (-23.206, 135.40) | (-25.492, 123.14) | (-22.132, 127.46) | (-26.262, 171.18) | (-26.236, 113.26) |
| | c | 0.3386 | NA | 0.0229 | 0.0020 | 0.2812 |

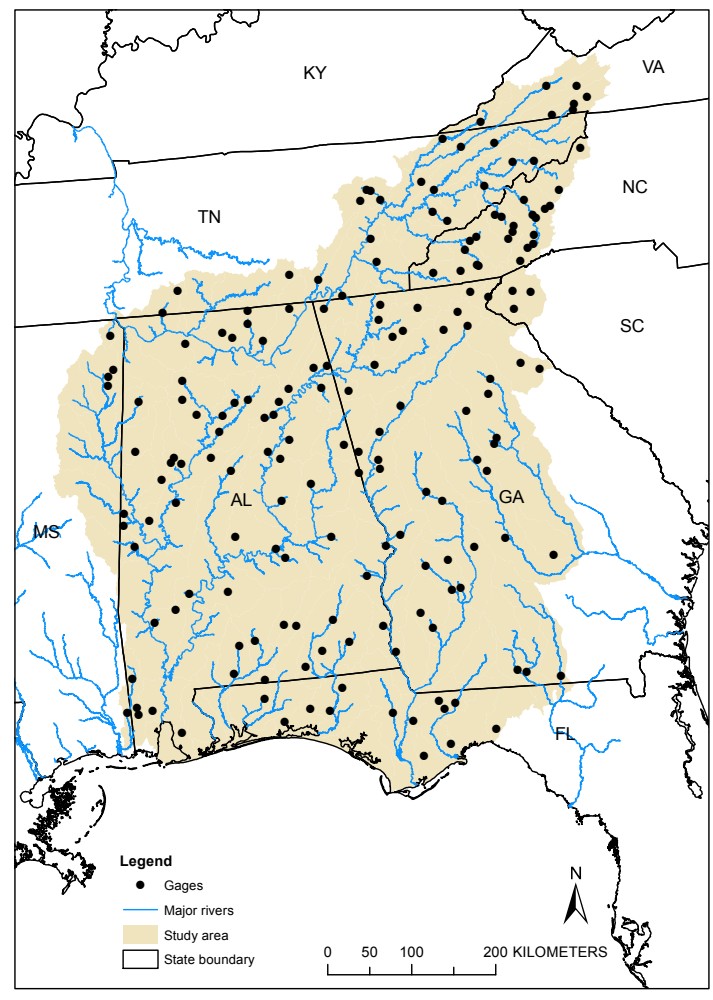

**Figure 1.** Map of study area showing the locations of 182 streamgages used for analysis and validation.

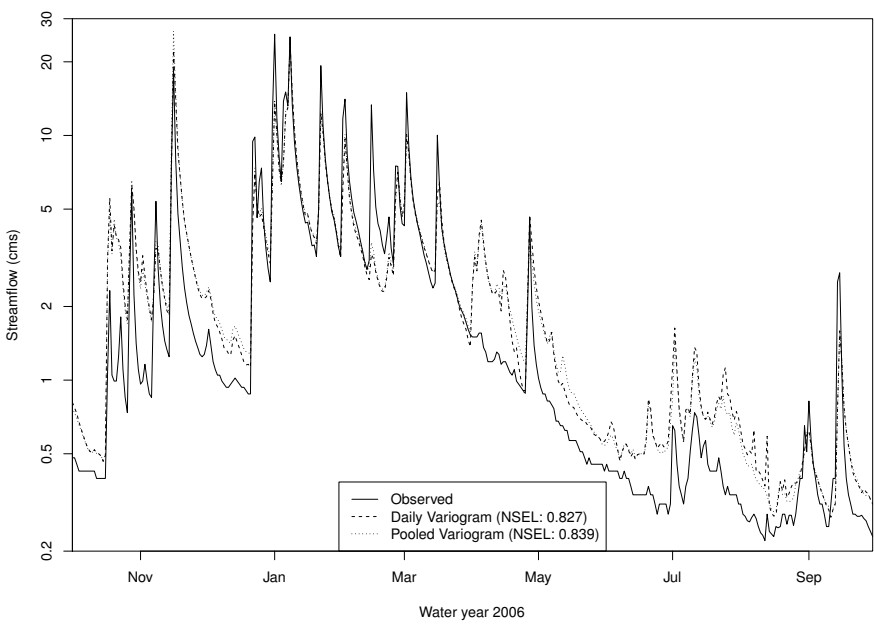

**Figure 2.** An example of the observed and simulated streamflows for a site and year selected to represent the median performance. The results are from site 02401390, with a drainage area of 365 $km^2$. Streamflow values are reported in cubic meters per second (cms).

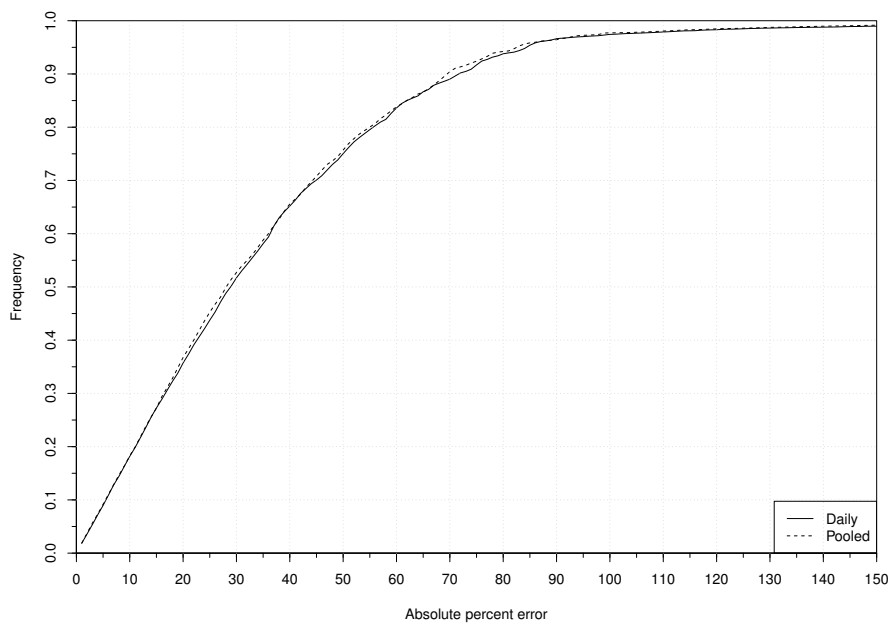

**Figure 3.** Median cumulative distribution of absolute percent errors in daily estimates for streamflow estimated from both daily and pooled variogram parameter sets.

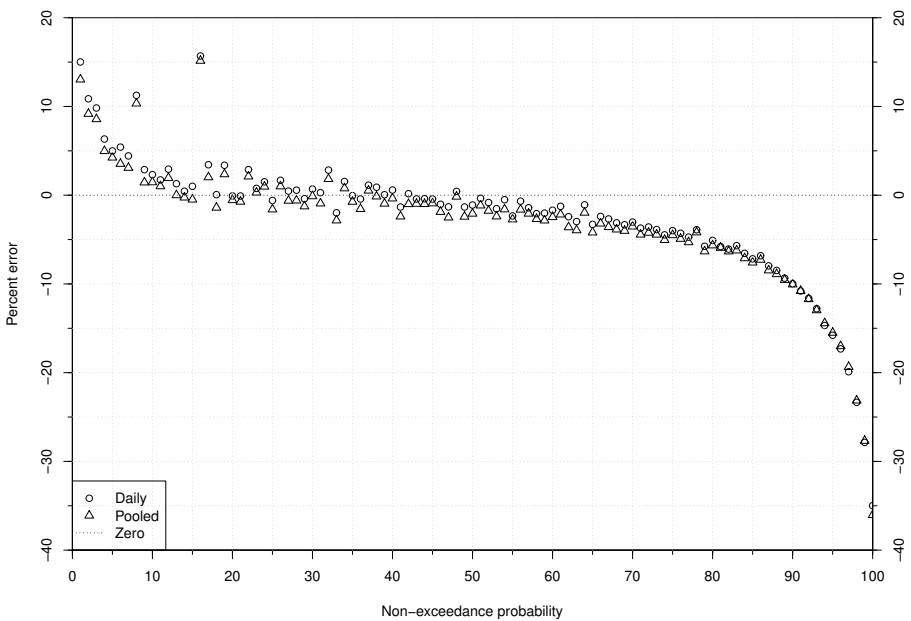

**Figure 4.** Median percent error for each non-exceedance probability, binned by single percentage points, for streamflow estimates from both daily and pooled variogram parameter sets.

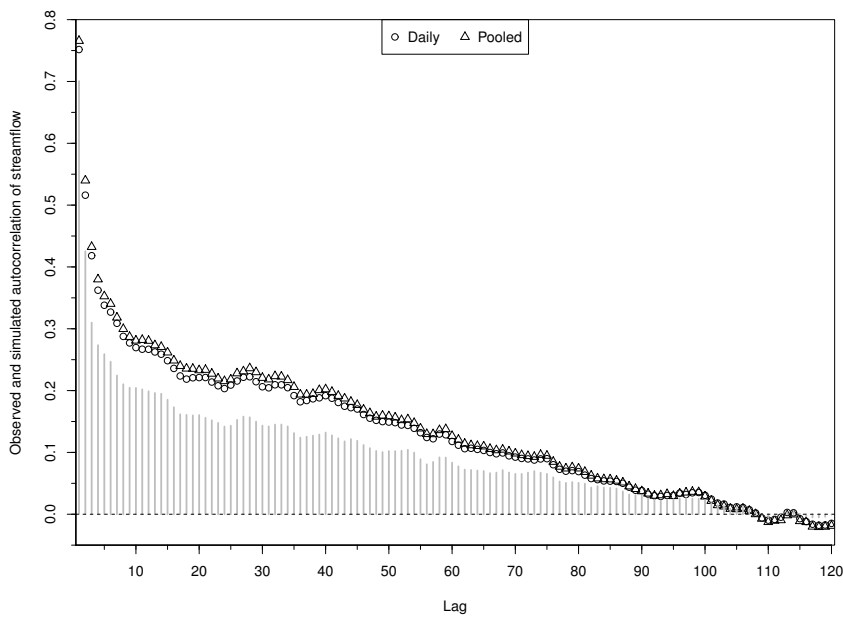

**Figure 5.** Observed autocorrelation of daily streamflow, in gray, with simulated autocorrelations from daily and pooled ordinary kriging daily streamflow time series.

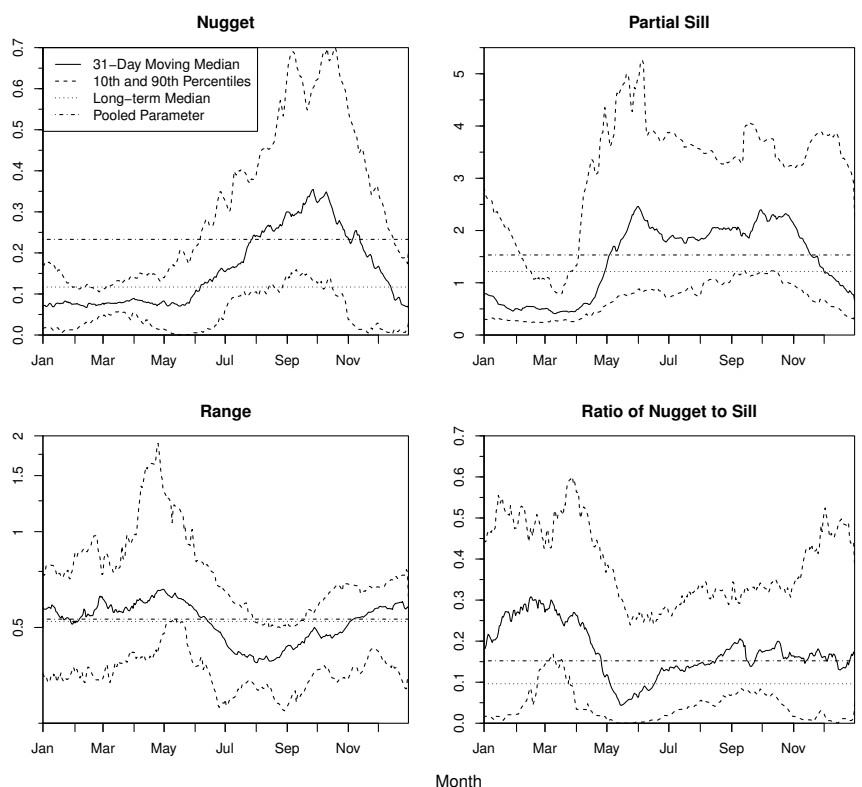

**Figure 6.** 31-day moving median of daily variogram parameters and the ratio of nugget to sill. (NOTE: The vertical axis of range is scaled by a factor of $10^6$.)