# Peer review of "Ordinary kriging as a tool to estimate historical daily streamflow records"

_Hydrology and Earth System Sciences, 2015_

## Referee Comment (RC1) · E. Pebesma (Referee) · 29 Feb 2016

article

Review of: *Ordinary kriging as a tool to estimate historical daily streamflow records*, by W. H. Farmer; reviewed by Edzer Pebesma.

Ordinary kriging is a well established method for spatial interpolation of geostatistical (field) variables. The current manuscript demonstrates its successful application to the interpolation of daily streamflow for ungauged catchments. I would be the last one to criticize the application of a simple, well-known technique to solve a clear problem in hydrology – hydrology has suffered enough of solutionism, papers defending new methods to solve problems where simpler alternatives would have been good enough. The current paper however leaves a number of questions open, related to understand-

ing why the proposed approach works well.

Physically, streamflow records are aggregates over larger regions: rain falls on the entire catchment, and much of it flows out in the stream. This implies that catchment size has an influence on the characteristics of the variable. Also, streamflow records may have been measured on the same river, introducing correlations because one gauge's catchment is contained by that of another. Ordinary kriging assumes intrinsic stationarity, and hence ignores catchment area and containment.

A recent variety of kriging, called top-kriging, has been developed (and is available as R package rtop, on CRAN) to accomodate variables with varying support, and was designed for this particular problem. In this paper, we see that ordinary kriging performs very similar (in terms of average statistics) to top-kriging. It would be good to better understand why the differences are so small: is it because we divide stream flow over catchment area? Is it because catchments don't contain each other? Is it because area is similar in most, or many cases? A graph of for instance the (temporal) variation of $z$ for each gauge against the size of the catchment might reveal a lot.

The author interprets the results as ordinary kriging (with pooled variogram) being favourable over top-kriging. I would consider them identical, as a difference of 1% in $R^2$ or RMSE is in my opinion meaningless for operational purposes. What interested me (but is not mentioned in the main text) is that the 90-percentile values for top-kriging perform better with a slightly larger margin. Is there an explanation for that?

I find the reporting of both $R^2$ and RMSE a bit artificial, in particular the fact that they give rise to different conclusions (I would have concluded that performance is similar). In (Pebesma, E.J., P. Switzer, K. Loague, 2005. Error analysis for the evaluation of model performance: Rainfall-runoff event time series data. Hydrological Processes, 19, 1529-1548, http://dx.doi.org/10.1002/hyp.5587) we point out that $R^2$ is a scaled version of MSE, so if RMSE gives a different ranking of methods than $R^2$, this is all due to taking the square root. If then, based on that, pooled ordinary kriging turns out to

be favoured, a good story why taking the square root of MSE is important would be no luxury!

The handling of zeroes: which fraction of the observation was zero? How sensitive were the results for the arbitrary number assigned to zero streamflows? (In my experience, when taking logs, this decision may pretty much blow away everything else).

Logarithms: is the goal to estimate $z$, i.e. not back-transform? If back-transformation was applied, how was this done? The results in Table 1 show results for non-log and log-transformed values, but the main text suggests that $z$, meaning only logarithms, are considered. What is the case?

Which software and software packages did you use to carry out this study? Can you also cite its authors? Since the data are open, can you provide a script that reproduces the findings?

The paper's main contribution seems how it handles time: instead of repeatedly computing and fitting variograms for each time step, a single (pooled) variogram model is fitted to the average of all variograms, and this seems to work better. As referenced in the paper, we (Gräler, Gerharz and Pebesma, 2011) found similar results when interpolating daily PM10 values over Europe. My impression there (and feeling here) is that the problem with daily fitted models is that occasionally the fit looks crazy, due to extreme values or strange conditions. This paper does not confirm nor deny this, as figure 6 only shows moving 31-day medians of variogram parameters. Can you also show (or describe) the time series of the raw daily values?

Given that figure 6 shows clear seasonal signals in the variogram parameters, an alternative to the current approach would be to use the 31-day median parameter values instead of the temporally constant pooled variogram model. This would be a compromise between the (too noisy?) daily fitted model and the (overly smooth?) constant model. Another question that might be discussed is the option to use spatio-temporal (ordinary, or top) kriging: right now, temporal correlation in streamflow records is ignored. In case the prediction would concern incidentally missing values, the observation directly before or after the missing value might be much more informative than the spatial neighbouring values. It might be the case that missingness means longer periods of no observations, in which case temporal correlation will not help much. Explaining the pattern of missing-ness might help the reader understand why this study did not consider temporal correlation; currently such an explanation is missing.

In general, the manuscript confuses semivariance with covariance; I strongly suggest to use only one of the two.

Textual Details:

1. if Figure 1 would show the (main) rivers, or catchments, we could see the degree to which catchments are contained by each other.

2. 4:21 $i = [1, ..., n]$

3. 4:21 Euclidian location: omit Euclidian

4. 5:9 then should be that

5. 5:9 $\mu$ is the Lagrange multiplier, not an estimate of the mean of $z$

6. 5:10 "In practice, the elements of D cannot be calculated explicitly" what you try to say is that individual covariance values cannot be inferred from a single sample (realisation) of $z$; only with additional stationarity assumptions, covariance can be modelled as a function of separation distance. Rephrase?

7. 5:12 a variogram is not a model of the covariance

8. 5:21 "theoretical variogram model", replace with "variogram model type".

9. 5:22 "in building the empirical variogram, the covariances": again, variogram/semivariance and covariogram/covariance are two different measures.

10. 5:23 for a complete description, the maximum distance up to which semivariances were computed is needed too (as well as whether the ten groups are of equal distance interval width)

11. 6:6 same confusion: the variogram does not have covariance values.

12. 6:15 "stationarity" → "temporal stationarity"

13. 6:20 "computation of as many variograms" → "fitting of as many variogram models"

14. I feel that it should be pointed out somewhere that averaging variogram model parameters does not necessarily lead to the same model as fitting a model to averaged (pooled) sample variograms.

15. 8:17 "and" → "a"

---

## Referee Comment (RC2) · J. O. Skøien (Referee) · 2 Mar 2016

This manuscript analyses ordinary kriging for estimation of historical daily streamflow. The paper is well written, and includes interesting analyses. Some revisions are still necessary before it can be published. Below are some suggestions for improvement.

There is some overlap between results and discussion, where some results are discussed in the results section, and then this is partly repeated in the discussion, which is then more like a summary. I think this part of the manuscript could be better organized.

Some work is based on the author's PhD (Farmer, 2015). I am not sure how easily accessible this PhD is? I think it is ok to include results from the PhD in this manuscript as previously unpublished, as long as they have not been presented in other peer re-

viewed journals/conference proceedings. The paragraph on P6 L12-17 should anyway be rewritten. I don't see why it is not intuitive to model each day independently? What is the most extreme? What is meant by stationarity of variogram parameters here, that they are temporally constant?

The brief summary of kriging is not brief; it pretty much includes everything in Skoien et al. (2006), which the author refers to, just in a different way. What is missing is actually a variogram model and the sample variogram, which is of more interest for the analyses than the equations for finding the weights. I think it is necessary to discuss in more detail the advantages/disadvantages of OK/TK and pooled/daily variograms beyond the cross-validation results. First of all, OK can be seen as the most extreme case of TK with only one regularization point, and comparing with the results of Skoien et al (2015), it is not surprising that also OK can perform reasonably well for most catchments. However, this is likely to depend on the configuration of observation and prediction locations. Table 1 indicates that the NSE of TK is considerably higher for the 10th percentile. Then the author barely mentions the prediction uncertainty, where I would expect daily variograms to perform better than pooled variograms, and TK should perform better than OK.

Regarding biases of upper/lower extremes, particularly P8 L3-10 and P12 11-14 – I think it is not surprising that kriging tends to over/underestimate the extremes. Kriging is generally unbiased around the 50th percentile, and a range around this percentile. For the conclusions, kriging is a stochastic approach, not deterministic, and using simulations would only create a range of values which is still centred around the under/overestimated prediction. Higher nugget effects will increase the tendency to smooth the extremes in the interpolated field. The average underestimation of 40% for the largest streamflows is still somewhat large, it would be useful to further analyze and discuss the causes for such large deviations, and see if there are particular cases where they are larger. On the other hand, the prediction difference might be smaller, as this refers to exceedance probability, something which could be better explained. It

is not easy to understand from the text what is meant by exceedance probability in P8. The sentence "For low streamflows, below . . ." is not clear either.

Although not frequently in use as far as I know, there are some methods for unbiased kriging of extreme values. Two of them are IWQSEL (Craigmile et al., 2006) and Modified Ordinary Kriging (Skoien et al, 2008), both methods implemented in the R-package intamap. I don't think this would be feasible to include for the analyses in this manuscript, but could be a possibility for future work. The results from the analyses of autocorrelation on P8 should probably not focus so much at what I would refer to as relative differences in percent (as is currently done, although referred to as absolute), rather the absolute errors. As mentioned, relative differences when autocorrelation is below 0.1 can be large, but still negligible. A figure could present the autocorrelation as lines.

The discussion about temporal variations of variogram parameters at the end of 3.2 should also include some more thoughts about the reason for the temporal changes, and check the description of the current relationships. I would assume that shorter ranges in Summer is an indication of more heterogeneity (more convective precipitation) and that long ranges is an indication of homogeneity. I would also assume that sill is decreasing in Winter because runoff is decreasing, so the ratio sill/mean runoff could be of interest, the same with the nugget/sill ratio. Long ranges in Winter/spring could be related to snow/snow melt. "beyond the range" is confusing.

Minor comments

P2 L12 "It is postulated" – Who postulated what?

P2 L27-28 There are also kriging methods where the predictions are based on external variables in addition to geolocation.

P3 L9 "Deterministic rainfall-runoff models" is more commonly used than mechanistic.

P3 L9-10 I would say that the comparison is missing from their work, they did not

emphasize the need.

P3 L25 logarithms of UNIT runoff?

P4 L1 The sentence is a bit clumsy (Because . . . because), consider rewriting, such as ". . . reference-quality in the designation (. . .) or in previous flood-frequency studies (. . .)." Remove brackets from next sentence.

P4 L10 One out of 33 days on average or some periods of 1-33 days? These were filled by the author or has previously been filled in by USGS?

P4 L16-17 This sentence seems unnecessary complicated, and I am not sure if it is completely correct.

P5 L17 Which previous hydrologic geostatistics is the OK in this manuscript an extension to? And depending on the answer, is OK really an extension or does the manuscript include analyses which are useful as an addition to other methods?

P5 L30-31 I think "the temporal considerations" can be deleted.

P6 L6 I don't think it is the covariance, but it could be the variance, due to short scale variability or measurement errors.

P6 L8 "the dependent variable" can be deleted, together with "of" (structure of which)?

P6 L21 What is stability of the spatial covariance structure?

P6 L23 Move first sentence to L25 (after daily variogram sentence).

P6 L31-32 Here it is a bit unclear what is meant by "average model". In L15-16 it is referred to averaging of model parameters, which is definitely different than a variogram model fitted to a temporally pooled empirical variogram. If the difference between average and pooled refers to the difference between treating each daily empirical variogram as equal, or giving them weights according to the number of pairs in each bin, then this should be described more explicit.

P7 L11 The usage of top-kriging in this manuscript is not exactly the same as the one described in Skoien and Bloschl (2007). The previous paper uses spatio-temporal regularization, whereas the implementation in rtop only uses spatial regularization. However, it can be assumed that the difference between these is small, and not likely to affect the quality of predictions. A formal comparison has not been done, but the current version of rtop uses a similar method as the one in the manuscript for predicting time series of runoff.

P7 L27 Is this poor performance in the recession period typical for all catchments, or is this example worse than many others? I would expect a bias in the extreme, as presented later in the paper, but also that some catchments will even be underestimated during May and June in the figure.

P7 L31 - P8 L2 I find this sentence somewhat unclear, in addition, is the difference significant? For me there is barely any difference between the curves. Regarding the figure, I usually don't like grids in figures, but this could be an exception where it might add some information.

P8 L17 and -> a

P11 L15-17 I did not understand this sentence. What is meant by hours and days per site?

P11 L18 This was correct in the past, but pooled variogram estimation is now included in the package, together with time series interpolation.

Craigmile, P. F., N. Cressie, T. J. Santner, and Y. Rao. 2006. A loss function approach to identifying environmental exceedances. Extremes, 8, 143-159.

Skoien, J. O., G. B. M. Heuvelink, and E. J. Pebesma. 2008. Unbiased block predictions and exceedance probabilities for environmental thresholds. In: J. Ortiz C. and X. Emery (eds). Proceedings of the eight international geostatistics congress. Gecamin, Santiago, Chile, pp. 831-840.

---

## Referee Comment (RC3) · A. Pugliese (Referee) · 18 Mar 2016

The paper "Ordinary kriging as a tool to estimate historical daily streamflow records" by Farmer W.H. shows a comparative assessment of kriging techniques, exploring the performances obtainable employing Ordinary Kriging, under different model settings, for the prediction of daily streamflow series in ungauged basins. The paper is well written and is rather complete in all its section, the topic is of wide interest in the hydrological field, thus I believe it is suitable for the publication in HESS after some little improvements that in my view the author should consider to take into account.

[Figure]

**Major comments**

1. Even if there is a relationship between the covariance function $C(x_1, x_2)$ between two data points $x_1$, $x_2$ and the variogram, which is, by definition: $\gamma(h) = 1/2E[(Z(x_2) - Z(x_1))^2]$, where $h = x_2 - x_1$ is the spatial Euclidean difference between the two data points and E is the expected value of the squared increment of $Z$, relative to the spatial lag $h$. All the textbooks and papers on geostatistics refer to the variogram, rather than employ the covariance directly, as the major controller of the spatial correlation. $C$ and $\gamma$ are two sides of the same coin, because $\gamma(h) = C(0) - C(h)$, though the variogram has some more features, which is why it is the main function to look at. For instance, most of the variables that might be referred to as "spatial fields" may (or may not) have a nugget effect, which is a unique feature of the variogram. Moreover, there are some variables that might be "non-stationary". In this case, one can denote non-stationarity as the variogram diverge and never reach the "sill", while non-stationarity might not be seen from the covariance. I think the mathematical notations and equations (1), (2), (3) and (4) (L25 P4) are formally incorrect as they refer to the covariance, rather they should refer to semivariances of the increment $z(x + h) - z(x)$, both theoretical or experimental (see for examples, Cressie, 1993; Journel and Huijbregts, 1978). Although there is a way to employ the covariance matrix too, which derives from the optimization of the prediction variance, the author did not report the correct one. I would recommend the author to rewrite the system of equation (2), (3) and (4) and stick with the variogram. Furthermore the author cites Skøien (2006) as the refernce for solving the kriging system. There are a couple of mistakes with this reference: (1) that paper focuses on solving an "adapted" ordinary kriging linear system to fit with regularized variorgams, so maybe this is not the best choice for someone who wants to discover more about kriging tecniques and (2) that paper never reports covariaces within matrices of the kriging linear system to solve, rather it reports correctly variograms.

2. I think that the comparison with Top-kriging here is not informative as it should and might be even misleading. Firstly, it does not report the best model setting. Even if the author specifies here and there that the comparison with Top-kriging is not definitive, I would strongly recommend to point out that Top-kriging model performances might not be the best obtainable in this study area. Or, in case preliminary analyses have shown that it is instead the best model setting, this should be clearly said throughout the manuscript. Secondly, I think the paper, which is intentionally unbalanced towards the two ordinary kriging methods, does not accomplish the assessment Top-kriging deserves. Indeed, the latter is actually an ordinary kriging too, technically it is a "modified" ordinary kriging, where the modification relies just on the variogram. The author instead groups this method together with DAR and QPPQ, whereas it should be grouped with the ordinary kriging methods. Concluding, does this comparison with Top-kriging reflect what the title says? At the very end, is this informative? Thus, should the Top-kriging be removed from the comparative assessments with other models?

**Minor comments**

1. L 24-25 P10. The author conclude that kriging techniques are biased and inaccurate in the tails of the distributions, and prove it with Fig. 4. This is somehow understandable and even quite normal. The kriging techniques are weighted average. Predicting streamflows within a leave-one-out cross-validation, when the lowest or highest streamflow is removed, plus it is perhaps orders of magnitude lower or higher than streamflows from donor sites, it is predictable that the outcome shows upward or downward biases, respectively, in those regimes. I think this thought might be taken into account, or at least pointed out clearly, maybe after those lines or elsewhere in the first sections.

2. In case the author adopted unconstrained kriging methods, that is when kriging weights are both positives and negatives, predicting low flows may lead to negative estimates. Would be interesting to know, if any, how many negatives are produced.

3. By taking the logs of the standardized streamflows, the author implicitly removes zero flows, if any, from the dataset. Is there any catchment with intermittent regime? If so, would be interesting to see how zero flows are treated.

4. L. 33 P9: it is not clear whether or not the author adopted the leave-one-out cross-validation for the DAR and QPPQ method too. I think so, but I would recommend to rephrase and be a little clearer about the cross-validation procedure used for all the methods reported. Would be even informative to know if any other cross-validation methods have been used in the past for DAR and QPPQ.

5. L. 31-32 P6. I think the parentheses might be removed and extending the text for a few lines might improve the reasoning.

**Technical notes and misspellings**

1. L. 32 P6, I've noticed the author use to put the punctuation mark before the right hand parenthesis, please correct throughout the manuscript with ")."

2. L2 P8. "[. . .] developed form", should be "developed from".

3. L8 P8. "[. . .] between the 5% and 15% non-exceedance probability" should be perhaps "between the 5% and 15% error"?

4. L30-31 P10. There are two "similarly" adverbs very close one to another. Please, consider to substitute one of them.

5. Fig. 4 height might be increased. In general Fig. 4 form factor might be changed to improve the readability of the figure itself.

**References**

Cressie, N.A.C., 1993. Statistics for spatial data, Wiley series in probability and mathematical statistics: Applied probability and statistics. J. Wiley.

Journel, A., Huijbregts, C., 1978. Mining Geostatistics. Acad. Press.
* * *

---

## Author Comment (AC1) · 5 Apr 2016

W.H. Farmer

wfarmer@usgs.gov

**Reviewer Comment 1:** *Ordinary kriging is a well established method for spatial interpolation of geostatistical (field) variables. The current manuscript demonstrates its successful application to the interpolation of daily streamflow for ungauged catchments. I would be the last one to criticize the application of a simple, well-known technique to solve a clear problem in hydrology – hydrology has suffered enough of solutionism, papers defending new methods to solve problems where simpler alternatives would have been good enough. The current paper however leaves a number of questions open, related to understanding why the proposed approach works well.*

**Author Response 1:** Thank you for your deep consideration of this work. I greatly appreciate your encouragement and am confident that your comments will greatly improve this manuscript. I hope that this work has interested you and motivated additional research.

**Reviewer Comment 2:** *Physically, streamflow records are aggregates over larger regions: rain falls on the entire catchment, and much of it flows out in the stream. This implies that catchment size has an influence on the characteristics of the variable. Also, streamflow records may have been measured on the same river, introducing correlations because one gauge's catchment is contained by that of another. Ordinary kriging assumes intrinsic stationarity, and hence ignores catchment area and containment.*

*A recent variety of kriging, called top-kriging, has been developed (and is available as R package rtop, on CRAN) to accomodate variables with varying support, and was designed for this particular problem. In this paper, we see that ordinary kriging performs very similar (in terms of average statistics) to top-kriging. It would be good to better understand why the differences are so small: is it because we divide stream flow over catchment area? Is it because catchments don't contain each other? Is it because area is similar in most, or many cases? A graph of for instance the (temporal) variation of z for each gauge against the size of the catchment might reveal a lot.*

**Author Response 2:** The reviewer raises two important concerns: the impact of drainage area and the impact of nested basins. Drainage area is incorporated into the estimation by considering depths of streamflow (line 18, page 5). This standardization removes some of the aggregating effects. Nested basins are not explicitly addressed. However, top-kriging does consider this effect. The minimal difference between ordinary and top-kriging suggests that the impact of nested basins is small.

The attached figure shows temporal variation (standard deviation of the logarithm of daily unit runoff) against catchment size. There is some variation with drainage area, but the Pearson correlation is weak (0.05). It appears that taking the logarithms on the unit runoff effectively controls for the effects of drainage area. This may be the reason for minimal improvements provided by top-kriging. This discussion, though not

the figure, will be added to the manuscript.

*Reviewer Comment 3: The author interprets the results as ordinary kriging (with pooled variogram) being favourable over top-kriging. I would consider them identical, as a difference of 1% in R2 or RMSE is in my opinion meaningless for operational purposes. What interested me (but is not mentioned in the main text) is that the 90-percentile values for top-kriging perform better with a slightly larger margin. Is there an explanation for that?*

**Author Response 3:** I strongly agree that the methods of kriging, ordinary and top, produce operationally identical performance. I tried to express this fact by noting the inconclusive advantage on lines 11-17 and 25-26 of page 10 and lines 9-21 of page 11 and lines 27-30 of page 12. The advantage of ordinary kriging, as you note earlier, is that it is simpler than top-kriging. If the difference is nearly identical, then the additional complexity and computational load of top-kriging may not be advantageous. I will attempt to make this understanding clearer in the noted lines.

As for the differing distributions of performance, I cannot explain this without a deeper exploration of the weakness of kriging methods, an exploration beyond the scope of this work. However, I will be sure to note this difference in the discussion of results. It could be that the advantages of top-kriging are significant at particular sites or across particular sub-regions, while they are un-impactful elsewhere. This might cause the already-poor sites to continue performing poorly, while another subset might improve significantly. Again, this discussion will be added to the manuscript.

*Reviewer Comment 4: I find the reporting of both R2 and RMSE a bit artificial, in particular the fact that they give rise to different conclusions (I would have concluded that performance is similar). In (Pebesma, E.J., P. Switzer, K. Loague, 2005. Error analysis for the evaluation of model performance: Rainfall-runoff event time series data. Hydrological Processes, 19, 1529-1548, http://dx.doi.org/10.1002/hyp.5587) we point out that R2 is a scaled version of MSE, so if RMSE gives a different ranking of methods*

*than R2, this is all due to taking the square root. If then, based on that, pooled ordinary kriging turns out to be favoured, a good story why taking the square root of MSE is important would be no luxury!*

**Author Response 4:** I agree that the performance metrics presented in Table 1 are not entirely independent of each other. In fact, they were all included for this very reason. The Nash-Sutcliffe model efficiency, as described by your reference and which I understand you to be calling R2, is a function of the mean squared error and is therefore related to the root mean squared error. However, the standardization by variance in the model efficiency can be shown to introduce confounding effects that lead to the differing conclusions of efficiency and root mean squared error. Gupta et al. (2009, DOI: 10.1016/j.jhydrol.2009.08.003 ; 2011, DOI: 10.1029/2011WR010962 ) show that the model efficiency is an amalgamation of the mean squared error and the Pearson correlation between observed and simulated response, among other things. For this very reason, both the Pearson correlation and root mean squared error are included in Table 1. The intent is to allow the reader to consider several common metrics and to tease out what is driving the Nash-Sutcliffe efficiency. The fact that root mean squared error and model efficiency do not agree on the significance of the differences between ordinary and top-kriging arises from the interplay between Pearson correlation and root mean squared error within the model efficiency. As I will point out in the revised manuscript, the decomposition allows one to better understand the components of the Nash-Sutcliffe model efficiency.

*Reviewer Comment 5: The handling of zeroes: which fraction of the observation was zero? How sensitive were the results for the arbitrary number assigned to zero streamflows? (In my experience, when taking logs, this decision may pretty much blow away everything else).*

**Author Response 5:** A full description of the data set used can be found in Farmer et al. (2014); their data was used without undocumented modification. Across the 182 streamgages considered, there were 1.6 million observations, of which 5,435 were

measured as zero. This is an average occurrence of 0.3% at each site. In fact, only 7 out of the 182 streamflow records considered contained days with zero-measured streamflow. Within these, the frequency of zeros is 0.99%, 0.06%, 31.2%, 14.6%, 4.00%, 4.09% and 1.42%. A statement on the prevalence zeros will be added to the manuscript.

The use of a placeholder value can certainly have a significant impact on the interpretation of the results. However, Farmer et al. (2014) found their results relatively insensitive to the selection of the placeholder because of the minimal fraction of zeros and zero-containing sites in the data set. However, if the zeros had been more prevalent or effected more sites, then it would certainly present a significant challenge. This discussion will be added to the manuscript.

*Reviewer Comment 6:* *Logarithms: is the goal to estimate z, i.e. not back-transform? If back-transformation was applied, how was this done? The results in Table 1 show results for non-log and log-transformed values, but the main text suggests that z, meaning only logarithms, are considered. What is the case?*

**Author Response 6:** The kriging system was designed to estimate the natural logarithm of unit runoff. However, to contextualize the results of estimations, some performance metrics were calculated on the back-transformed estimates. This back-transformation was done using simple exponentiation rather than developing a unique bias correction factor. The development of a bias correction factor that could be applied in the case of an ungauged basin was beyond the scope of this manuscript but is surely interesting to future research.

It should also be noted that the skewed distribution of observed daily streamflow may negatively affect the interpretation of the Nash-Sutcliffe model efficiency of streamflow (Gupta et al., 2009, DOI: 10.1016/j.jhydrol.2009.08.003 ). For this reason, the Nash-Sutcliffe of the logarithms of streamflow is the more reliable metric. This fact further obscures the need for an explicit bias correction in this application. In addition

to the previous discussion of performance metrics, this discussion will be added to the manuscript.

*Reviewer Comment 7: Which software and software packages did you use to carry out this study? Can you also cite its authors? Since the data are open, can you provide a script that reproduces the findings?*

**Author Response 7:** This work relied on the geoR package developed by Ribeiro and Diggle (2015, geoR: Analysis of Geostatistical Data. R package version 1.7-5.1. http://CRAN.R-project.org/package=geoR). The scripts required to produce this data are not currently available in a publishable format nor do I have the capacity to revise them to a publishable format at this time. However, I do think that a pooled estimation procedure would be a great addition to packages like geoR and gstat. For future work, I would be happy to discuss this development with you further.

*Reviewer Comment 8: The paper's main contribution seems how it handles time: instead of repeatedly computing and fitting variograms for each time step, a single (pooled) variogram model is fitted to the average of all variograms, and this seems to work better. As referenced in the paper, we (Gräler, Gerharz and Pebesma, 2011) found similar results when interpolating daily PM10 values over Europe. My impression there (and feeling here) is that the problem with daily fitted models is that occasionally the fit looks crazy, due to extreme values or strange conditions. This paper does not confirm nor deny this, as figure 6 only shows moving 31-day medians of variogram parameters. Can you also show (or describe) the time series of the raw daily values?*

**Author Response 8:** The daily parameter values do indeed appear crazy. This was why the 31-day moving average was used improve the readability of figure 6. I will add a description of this variability, noting that it was chaotic, to the paragraph in lines 23-28 of page 8. I will also include this in the discussion in lines 27-33 of page 10.

*Reviewer Comment 9: Given that figure 6 shows clear seasonal signals in the variogram parameters, an alternative to the current approach would be to use the 31-day*

*median parameter values instead of the temporally constant pooled variogram model. This would be a compromise between the (too noisy?) daily fitted model and the (overly smooth?) constant model.*

**Author Response 9:** It may be that there is a particular averaging procedure that optimizes model performance. This manuscript considered only two end-members of the continuum: daily parameters and time-invariant parameters. Using a moving-average parameter set may dampen daily effects and improve performance but may also reduce the computational advantages of time-invariant models. The exploration of the continuum will be added to the revised manuscript, but is beyond the scope of this work. Certainly, the results herein encourage exploration of alternative time-averaging.

*Reviewer Comment 10: Another question that might be discussed is the option to use spatio-temporal (ordinary, or top) kriging: right now, temporal correlation in streamflow records is ignored. In case the prediction would concern incidentally missing values, the observation directly before or after the missing value might be much more informative than the spatial neighbouring values. It might be the case that missingness means longer periods of no observations, in which case temporal correlation will not help much. Explaining the pattern of missing-ness might help the reader understand why this study did not consider temporal correlation; currently such an explanation is missing.*

**Author Response 10:** Spatio-temporal kriging presents another opportunity for explicitly handling the temporal variation of streamflows and variogram parameters. As the reviewer suggests, this would be especially important for applications seeking to fill-in sparse records. Here, the question of temporally sparse records was not considered. For purposes of validation, each site was treated as if it were completely ungauged. Furthermore, the work of Skøien and Blöschl (2007, DOI: 10.1029/2006WR005760 ) showed only limited returns for spatio-temporal considerations in hydrologic time series applications. I will note this in the revised manuscript, but the exploration is left for future work.

***Reviewer Comment 11:*** *In general, the manuscript confuses semivariance with covariance; I strongly suggest to use only one of the two.*

**Author Response 11:** I will revise the manuscript to more clearly distinguish between covariance and semivariance.

***Reviewer Comment 12:*** *if Figure 1 would show the (main) rivers, or catchments, we could see the degree to which catchments are contained by each other.*

**Author Response 12:** These will be added to the figure.

***Reviewer Comment 13:*** *4:21 i = [1, ..., n]*

**Author Response 13:** Revised.

***Reviewer Comment 14:*** *4:21 Euclidian location: omit Euclidian*

**Author Response 14:** Revised.

***Reviewer Comment 15:*** *5:9 then should be that*

**Author Response 15:** Revised.

***Reviewer Comment 16:*** *5:9 $\mu$ is the Lagrange multiplier, not an estimate of the mean of z*

**Author Response 16:** Revised to read ". . .and estimates the LaGrange multiplier, $\mu$, to control for the unknown mean of z."

***Reviewer Comment 17:*** *5:10 "In practice, the elements of D cannot be calculated explicitly" what you try to say is that individual covariance values cannot be inferred from a single sample (realisation) of z; only with additional stationarity assumptions, covariance can be modelled as a function of separation distance. Rephrase?*

**Author Response 17:** The text in lines 10-14 of page 5 will be revised to read "The single realization of [C] that is produced from the sample observations of z cannot be considered to represent the underlying system. The sample may produce a matrix

that is singular or not positive definite, conditions required for solution of the system. Furthermore, the elements of [D], by nature, are unobservable as the value of the dependent variable at the ungauged location is what is being estimated. However, with additional assumptions of stationarity, semivariance can be modeled as a function of separation distance. Several classical models are available to ensure positive definiteness. These models are parameterized by calibration to the empirical variogram of observed semivariance as a function of distance. Once a variogram model is selected, the system becomes . . ."

*Reviewer Comment 18: 5:12 a variogram is not a model of the covariance*

**Author Response 18:** Revised to "using a model of semivariance". Per recommendations of other reviewers, this portion will be revised to present the kriging system as a function of semivariance alone. This will improve the confusion between covariance and semivariance.

*Reviewer Comment 19: 5:21 "theoretical variogram model", replace with "variogram model type".*

**Author Response 19:** Revised.

*Reviewer Comment 20: 5:22 "in building the empirical variogram, the covariances": again, variogram/semivariance and covariogram/covariance are two different measures.*

**Author Response 20:** "Covariances" will be replaced with "semivariances".

*Reviewer Comment 21: 5:23 for a complete description, the maximum distance up to which semivariances were computed is needed too (as well as whether the ten groups are of equal distance interval width)*

**Author Response 21:** The maximum distance was the maximum inter-site distance in the network, 920 kilometers. The bins were equal-interval bins. The following statement will be added: "... were stratified into ten equal-interval groups based on the

inter-site distances ranging from zero to the maximum inter-site distance of 920 kilometers, as suggested by ..."

*Reviewer Comment 22: 6:6 same confusion: the variogram does not have covariance values.*

**Author Response 22:** Replaced "covariance" with "semivariance".

*Reviewer Comment 23: 6:15 "stationarity" -> "temporal stationarity"*

**Author Response 23:** Revised.

*Reviewer Comment 24: 6:20 "computation of as many variograms" -> "fitting of as many variogram models"*

**Author Response 24:** Revised.

*Reviewer Comment 25: I feel that it should be pointed out somewhere that averaging variogram model parameters does not necessarily lead to the same model as fitting a model to averaged (pooled) sample variograms.*

**Author Response 25:** Per the additional recommendations of other reviewers, the discussion in the last paragraph of page 6 will be expanded to further clarify the distinction between the pooled model and the averaged model.

*Reviewer Comment 26: 8:17 "and" -> "a"*

**Author Response 26:** Revised.

[Figure]

**Variability of Unit Runoff as a Function of Drainage Area**

[Figure]

**Fig. 1.** Temporal variability in streamflow with drainage area.

---

## Author Comment (AC2) · 5 Apr 2016

**Reviewer Comment 1:** This manuscript analyses ordinary kriging for estimation of historical daily streamflow. The paper is well written, and includes interesting analyses. Some revisions are still necessary before it can be published. Below are some suggestions for improvement.

Author Response 1: Thank you for a thorough and valuable review of my work. I greatly appreciate the effort you have put into improving the impact and communication of my work. I hope that, with your improvements, this work will motivate future research.

**Reviewer Comment 2:** There is some overlap between results and discussion, where some results are discussed in the results section, and then this is partly repeated in the discussion, which is then more like a summary. I think this part of the manuscript Printer-friendly version

could be better organized.

**Author Response 2:** I will revise the sections so that the 'Results' section is re-named as a 'Results and Discussion'. The 'Discussion' section will be revised and re-named 'General Discussion' as a subsection of 'Results and Discussion'.

**Reviewer Comment 3:** Some work is based on the author's PhD (Farmer, 2015). I am not sure how easily accessible this PhD is? I think it is ok to include results from the PhD in this manuscript as previously unpublished, as long as they have not been presented in other peer reviewed journals/conference proceedings. The paragraph on P6 L12-17 should anyway be rewritten. I don't see why it is not intuitive to model each day independently? What is the most extreme? What is meant by stationarity of variogram parameters here, that they are temporally constant?

**Author Response 3:** My Ph.D. dissertation is available through my alma mater, but is not freely-available online. Several chapters have been published elsewhere, but the chapter pertaining to this material has not been published.

I will revise the noted paragraph for clarity. Modeling days independently is non-intuitive because our basic understanding of hydrology shows strong temporal dependence across days. I called the averaging of parameters the most extreme case because it exists on a continuum of estimating every day and taking an average of all days. As discussed with a previous reviewer, you could consider any range of averaging (e.g., a 31-day moving average). I will revise to make this point more clearly. Here I am referring to temporal stationarity and will add that adjective.

**Reviewer Comment 4:** The brief summary of kriging is not brief; it pretty much includes everything in Skøien et al. (2006), which the author refers to, just in a different way. What is missing is actually a variogram model and the sample variogram, which is of more interest for the analyses than the equations for finding the weights.

Author Response 4: The reviewer is correct, even though the spherical variogram was
used (line 20, page 5), I could more clearly describe the variogram model considered. Per recommendations from another reviewer, I will revise the kriging system presented to be based on semivariance. I will also add the following description of the spherical variogram model:

$$\gamma(h) = \frac{1}{2}E[(Z(\mathbf{x}+h) - Z(\mathbf{x}))^2]$$
(1)

Where x is a geospatial location and *h* is a separation distance. To improve stability of the system, the semivariance  $\gamma(h)$  is approximated as

$$\hat{\gamma}(h) = \begin{cases} (\sigma^2 - \tau^2)(\frac{3h}{2\phi} - \frac{h^3}{2\phi^3}) + \tau^2 & if \quad h \le \phi \\ \sigma^2 & if \quad h > \phi \end{cases}$$
(2)

Where  $\sigma^2$  is the sill,  $\phi$  is the range and  $\tau^2$  is the nugget variance.

**Reviewer Comment 5:** I think it is necessary to discuss in more detail the advantages/disadvantages of OK/TK and pooled/daily variograms beyond the cross-validation results. First of all, OK can be seen as the most extreme case of TK with only one regularization point, and comparing with the results of Skøien et al (2015), it is not surprising that also OK can perform reasonably well for most catchments. However, this is likely to depend on the configuration of observation and prediction locations. Table 1 indicates that the NSE of TK is considerably higher for the 10th percentile. Then the author barely mentions the prediction uncertainty, where I would expect daily variograms to perform better than pooled variograms, and TK should perform better than OK.

**Author Response 5:** I agree that this is far from a thorough comparison of ordinary and top kriging. The intention of this paper is to introduce ordinary kriging as a competitor to standard methods. A more robust comparison, as was discussed with another reviewer, may be more appropriate in future research. There are several unanswered questions, including the configuration of observation and prediction locations. Why the 10th percentile of TK is better is not currently known. I will add text to point out this.
In line with the recommendations of another reviewer, I will add some prose to discuss the similarity of ordinary and top kriging. It is especially important to note that one is a modification of the other.

One of the main advantages of kriging methods is the availability of prediction uncertainty. However, the possibility for developing confidence intervals or uncertainty estimates on daily estimates is a subject that merits its own thorough research. As there is no standard method for single-index techniques, there are no grounds for comparison. Further research is needed, but I will note this in the manuscript.

**Reviewer Comment 6:** Regarding biases of upper/lower extremes, particularly P8 L3-10 and P12 11-14 – I think it is not surprising that kriging tends to over/underestimate the extremes. Kriging is generally unbiased around the 50th percentile, and a range around this percentile. For the conclusions, kriging is a stochastic approach, not deterministic, and using simulations would only create a range of values which is still centred around the under/overestimated prediction. Higher nugget effects will increase the tendency to smooth the extremes in the interpolated field.

**Author Response 6:** I referred to kriging as a deterministic approach because it produces a single estimate of daily streamflow from a single, deterministic input. I will clarify in the text. Of course, as you suggest and as I mention in the manuscript, the estimates can be coupled with prediction uncertainties to be used in a deterministic manner. However, the simulations will remain clustered around biased averages. What is needed is a bias correction factor. The derivation of such is beyond the scope of this work but is of immense interest and I look forward to future research on the topic.

**Reviewer Comment 7:** The average underestimation of 40% for the largest streamflows is still somewhat large, it would be useful to further analyze and discuss the causes for such large deviations, and see if there are particular cases where they are larger. On the other hand, the prediction difference might be smaller, as this refers to exceedance probability, something which could be better explained. It is not easy to HESSD
understand from the text what is meant by exceedance probability in P8. The sentence "For low streamflows, below . . ." is not clear either.

**Author Response 7:** I will clarify the text to refer to non-exceedance probabilities rather than percentiles. The 40% error is in high streamflows, making it all the more concerning. Why such large deviations occur is not well understood. Kriging, in of itself, is a smoothing estimator and the temporal pooling further smooths results. This smoothing may mask the effects of rare, extreme events and dampen their amplitudes. I will note this in the manuscript.

**Reviewer Comment 8:** Although not frequently in use as far as I know, there are some methods for unbiased kriging of extreme values. Two of them are IWQSEL (Craigmile et al., 2006) and Modified Ordinary Kriging (Skøien et al, 2008), both methods implemented in the Rpackage intamap. I don't think this would be feasible to include for the analyses in this manuscript, but could be a possibility for future work.

Author Response 8: I am excited to consider these methods in future work.

**Reviewer Comment 9:** The results from the analyses of autocorrelation on P8 should probably not focus so much at what I would refer to as relative differences in percent (as is currently done, although referred to as absolute), rather the absolute errors. As mentioned, relative differences when autocorrelation is below 0.1 can be large, but still negligible. A figure could present the autocorrelation as lines.

**Author Response 9:** I will consider revising this figure, as the reviewer suggests. I agree that is could be improved by showing the deviations in some way that minimizes the impact of inconsequential differences.

**Reviewer Comment 10:** The discussion about temporal variations of variogram parameters at the end of 3.2 should also include some more thoughts about the reason for the temporal changes, and check the description of the current relationships. I would assume that shorter ranges in Summer is an indication of more heterogeneity

**HESSD**
(more convective precipitation) and that long ranges is an indication of homogeneity. I would also assume that sill is decreasing in Winter because runoff is decreasing, so the ratio sill/mean runoff could be of interest, the same with the nugget/sill ratio. Long ranges in Winter/spring could be related to snow/snow melt. "beyond the range" is confusing.

Author Response 10: As the reviewer suggests, an understanding of the impacts of any parameter changing in isolation is not easily described. I chose to plot the raw parameters because they are constant across all sites when the whole network is used for calibration. Standardizing the sill by mean values thus becomes difficult. Though, I do agree that it would be useful for additional analysis. The ratio of nugget to sill is also useful.

As described on line 15 of page 9, I agree that the decreased range in Summer is emblematic of less homogeneity and more heterogeneity. I'll make this point more clearly. I will also add some of the reviewers' comments to the manuscript after exploring the ratio of sill to nugget. The attached figure shows the 31-day average ratio of nugget to sill. January through April, the nugget accounts for 20-30% of the sill, it dips to only 5% of the sill in mid-May and then steadies to about 15% of the sill through the rest of the year. The pooled parameter sets the nugget at 15% of the sill. Lower Winter sills may be the result of smaller streamflows. The measurement uncertainty in smaller streamflows may be greater, increasing the nugget. For high streamflows in mid-May, the measurement error represented by the nugget may be minimal. Average flows throughout the remainder of the year show a standard degree of measurement error, agreeing with the pooled parameter. I will consider adding this figure and discussion to the manuscript.

Reviewer Comment 11: P2 L12 "It is postulated" – Who postulated what?

Author Response 11: This sentence is meant to state my hypothesis that "predictions of daily streamflow time series can be improved by incorporating regional information
beyond the information available at single index streamgages and that, building on previous hydrologic time series analysis, this can be achieved by utilizing the geostatistical method known as kriging." I will revise to more clearly state my hypothesis by stating "It is hypothesized here that..."

**Reviewer Comment 12:** P2 L27-28 There are also kriging methods where the predictions are based on external variables in addition to geolocation.

**Author Response 12:** There are; I will revise the text to say that co-kriging is a variation of kriging that incorporates the influence of other variables.

**Reviewer Comment 13:** P3 L9 "Deterministic rainfall-runoff models" is more commonly used than mechanistic.

Author Response 13: I will change the term throughout.

**Reviewer Comment 14:** P3 L9-10 I would say that the comparison is missing from their work, they did not emphasize the need.

Author Response 14: I meant to imply that the lack of a comparison emphasizes the need, not that they emphasized the need. I will revise to "Because it has not been previously considered, it is important to explore and contrast the potential of ordinary kriging and top-kriging to predict streamflow time series in ungauged basins."

Reviewer Comment 15: P3 L25 logarithms of UNIT runoff?

Author Response 15: Yes, runoff is considered as a depth here, whereas discharge is considered a volume. I will add "unit" for clarity.

**Reviewer Comment 16:** P4 L1 The sentence is a bit clumsy (Because . . . because), consider rewriting, such as ". . . reference-quality in the designation (. . .) or in previous flood-frequency studies (. . .)." Remove brackets from next sentence.

Author Response 16: Will be revised as "...reference quality according to their designation ...". The next sentence will be removed from parentheses.

HESSD
**Reviewer Comment 17:** P4 L10 One out of 33 days on average or some periods of 1-33 days? These were filled by the author or has previously been filled in by USGS?

**Author Response 17:** These were previous filled by Farmer et al. (2014) and contained periods ranging in length from one to 33 days. "on the order of one to 33 days" will be revised to "for periods of one to 33 days long".

**Reviewer Comment 18:** P4 L16-17 This sentence seems unnecessary complicated, and I am not sure if it is completely correct.

Author Response 18: Revised to "The inter-site semivariances of data from a measured network can be used to create a system of linear equations predicting the semivariance at an unmeasured site to be a linear sum of the semivariance between all observed sites."

**Reviewer Comment 19:** P5 L17 Which previous hydrologic geostatistics is the OK in this manuscript an extension to? And depending on the answer, is OK really an extension or does the manuscript include analyses which are useful as an addition to other methods?

**Author Response 19:** The application, not OK itself, is an extension of previous applications. The extension is that, here, time series are being considered rather than streamflow statistics. As the reviewer points out, this method is really an addition rather than an extension. I will revise to "Ordinary kriging of streamflow time series builds off of previous hydrologic applications to predict streamflow statistics to produce a method for handling temporal variation along with spatial variation."

**Reviewer Comment 20:** P5 L30-31 I think "the temporal considerations" can be deleted.

**Author Response 20:** Will revise to "focused on kriging time series and the temporal behavior of kriging parameters."

Reviewer Comment 21: P6 L6 I don't think it is the covariance, but it could be the
variance, due to short scale variability or measurement errors.

Author Response 21: Per the recommendation of another reviewer, I will revise to speak only in terms of semivariance. The nugget, therefore, is the semivariance of colocated points, which can be non-zero due to short-scale variability or measurement errors. Revisions will reflect this understanding.

**Reviewer Comment 22:** P6 L8 "the dependent variable" can be deleted, together with "of" (structure of which)?

Author Response 22: Will be revised to "In some previous hydrologic applications of kriging, the semivariance, which is modeled by the semivariogram, has been assumed to be temporally constant..."

Reviewer Comment 23: P6 L21 What is stability of the spatial covariance structure?

**Author Response 23:** Stability is meant to imply that the parameters of the semivariogram are constant. Will revise to "If the parameters of the semivariogram can be reasonably assumed to be constant, then the computational ..."

**Reviewer Comment 24:** P6 L23 Move first sentence to L25 (after daily variogram sentence).

Author Response 24: Revised.

**Reviewer Comment 25:** P6 L31-32 Here it is a bit unclear what is meant by "average model". In L15-16 it is referred to averaging of model parameters, which is definitely different than a variogram model fitted to a temporally pooled empirical variogram. If the difference between average and pooled refers to the difference between treating each daily empirical variogram as equal, or giving them weights according to the number of pairs in each bin, then this should be described more explicit.

Author Response 25: The reviewer's understanding is correct. Per additional advice from other reviewers, I will revise to more clearly state the difference between the

HESSD
average and the pooled model. "The average model treats each empirical variogram equally, while the pooled model weights each bin by the number of pairs in each bin."

**Reviewer Comment 26:** P7 L11 The usage of top-kriging in this manuscript is not exactly the same as the one described in Skøien and Blöschl (2007). The previous paper uses spatio-temporal regularization, whereas the implementation in rtop only uses spatial regularization. However, it can be assumed that the difference between these is small, and not likely to affect the quality of predictions. A formal comparison has not been done, but the current version of rtop uses a similar method as the one in the manuscript for predicting time series of runoff.

**Author Response 26:** Thank you for this information. I will add the statement "Using the same metrics, ordinary kriging was contrasted with an application of top-kriging similar to that defined by Skøien and Blöschl (2007). Top-kriging was applied using the rtop package (Skøien, 2015), which uses spatial regularization rather than the spatio-temporal regularization presented by Skøien and Blöschl (2007). The differences can be assumed to be negligible for this application. Regardless, here, top-kriging was applied..."

**Reviewer Comment 27:** P7 L27 Is this poor performance in the recession period typical for all catchments, or is this example worse than many others? I would expect a bias in the extreme, as presented later in the paper, but also that some catchments will even be underestimated during May and June in the figure.

**Author Response 27:** As can be seen in the figure, some recessions are reproduced well, while others are misrepresented. See, for example, the recessions from January through March and compare with the recession May through July. This variability in recession reproduction is typical; there is no categorical reproduction. Future research may explore the reproduction of specific streamflow regimes or signatures.

**Reviewer Comment 28:** P7 L31 - P8 L2 I find this sentence somewhat unclear, in addition, is the difference significant? For me there is barely any difference between
the curves. Regarding the figure, I usually don't like grids in figures, but this could be an exception where it might add some information.

Author Response 28: I will revise to note the insignificance of these differences. It may read "Though the differences between the curves from the pooled and daily variograms are not significant, the pooled variogram produces estimates with slightly fewer large errors." I will add grids to the figure.

Reviewer Comment 29: P8 L17 and -> a

Author Response 29: Revised.

**Reviewer Comment 30:** P11 L15-17 I did not understand this sentence. What is meant by hours and days per site?

**Author Response 30:** I will revise to "... (depending on processor speeds, top-kriging required just less than three days of computation time for each site predicted, while ordinary kriging required only hours of computation time per site predicted)."

**Reviewer Comment 31:** P11 L18 This was correct in the past, but pooled variogram estimation is now included in the package, together with time series interpolation.

**Author Response 31:** Will revise to "At the time of application, there package by Skøien (2015) did not contain a method to estimate pooled variograms directly. More recent versions do contain this functionality."

**HESSD**
Interactive

comment

31-day Moving Median of the Ratio of Nugget to Sill

---

## Author Comment (AC3) · 5 Apr 2016

*Reviewer Comment 1: The paper "Ordinary kriging as a tool to estimate historical daily streamflow records" by Farmer W.H. shows a comparative assessment of kriging techniques, exploring the performances obtainable employing Ordinary Kriging, under different model settings, for the prediction of daily streamflow series in ungauged basins. The paper is well written and is rather complete in all its section, the topic is of wide interest in the hydrological field, thus I believe it is suitable for the publication in HESS after some little improvements that in my view the author should consider to take into account.*

**Author Response 1:** I wish to thank the reviewer for his close attention to the details of this work. I very much appreciate the effort that was clearly put into this review and

think it will significantly improve the impact and communication of this work. I look forward to future discussions.

*Reviewer Comment 2: Even if there is a relationship between the covariance function C(x1, x2) between two data points x1, x2 and the variogram, which is, by definition: (h) = 1/2E[(Z(x2)−Z(x1))2] , where h = x2−x1 is the spatial Euclidean difference between the two data points and E is the expected value of the squared increment of Z, relative to the spatial lag h. All the textbooks and papers on geostatistics refer to the variogram, rather than employ the covariance directly, as the major controller of the spatial correlation. C and are two sides of the same coin, because (h) = C(0)−C(h), though the variogram has some more features, which is why it is the main function to look at. For instance, most of the variables that might be referred to as "spatial fields" may (or may not) have a nugget effect, which is a unique feature of the variogram. Moreover, there are some variables that might be "non-stationary". In this case, one can denote non-stationarity as the variogram diverge and never reach the "sill", while non-stationarity might not be seen from the covariance. I think the mathematical notations and equations (1), (2), (3) and (4) (L25 P4) are formally incorrect as they refer to the covariance, rather they should refer to semivariances of the increment z(x + h) − z(x), both theoretical or experimental (see for examples, Cressie, 1993; Journel and Huijbregts, 1978). Although there is a way to employ the covariance matrix too, which derives from the optimization of the prediction variance, the author did not report the correct one. I would recommend the author to rewrite the system of equation (2), (3) and (4) and stick with the variogram. Furthermore the author cites Skøien (2006) as the refernce for solving the kriging system. There are a couple of mistakes with this reference: (1) that paper focuses on solving an "adapted" ordinary kriging linear system to fit with regularized variorgams, so maybe this is not the best choice for someone who wants to discover more about kriging tecniques and (2) that paper never reports covariaces within matrices of the kriging linear system to solve, rather it reports correctly variograms.*

**Author Response 2:** The reviewer is correct that kriging systems are typically presented in terms of the semivariogram. However, for spatially stationary processes, as is assumed here, the kriging system defined in terms of covariances and semivariances results in identical weights. It is therefore, not formally incorrect to present the system in terms of covariances. The only advantage of considering semivariances explicitly is in the case on nonstationary processes. (The covariance structure still retains the nugget effect, which the reviewer notes as $C(0)$.) Furthermore, while Skøien et al (2006) presents a modified kriging system, their introduction of basic kriging is sufficient; I will add additional references. To ensure consistency with other hydrological applications, the kriging system will be rewritten to be framed in terms of the semivariance.

*Reviewer Comment 3: I think that the comparison with Top-kriging here is not informative as it should and might be even misleading. Firstly, it does not report the best model setting. Even if the author specifies here and there that the comparison with Top-kriging is not definitive, I would strongly recommend to point out that Top-kriging model performances might not be the best obtainable in this study area. Or, in case preliminary analyses have shown that it is instead the best model setting, this should be clearly said throughout the manuscript.*

**Author Response 3:** I agree with the reviewer that this is not an exhaustive comparison of ordinary and top kriging. With a more targeted exploration and application, it may indeed be possible to improve either model. I will certainly add some verbiage to document the limitations of this comparison.

*Reviewer Comment 4: Secondly, I think the paper, which is intentionally unbalanced towards the two ordinary kriging methods, does not accomplish the assessment Top-kriging deserves. Indeed, the latter is actually an ordinary kriging too, technically it is a "modified" ordinary kriging, where the modification relies just on the variogram. The author instead groups this method together with DAR and QPPQ, whereas it should be grouped with the ordinary kriging methods. Concluding, does this comparison with*

*Top-kriging reflect what the title says? At the very end, is this informative? Thus, should the Top-kriging be removed from the comparative assessments with other models?*

**Author Response 4:** The reviewer raises an interesting point, suggesting that it might be appropriate to exclude top-kriging from the comparison. The reviewer is correct that top-kriging is a modification of ordinary kriging rather than a completely different method. While I agree with the spirit of the reviewer's comment, I do feel that it is important to contextualize the ordinary-kriging result in the realm of other hydrologic kriging applications, most of which employ top-kriging. Furthermore, the inclusion of a basic comparison with top-kriging shows that modification to the ordinary kriging scheme only produces marginal improvements. I will revise the text to make this last more point more clearly and to point out that top-kriging is more akin to ordinary kriging than any other method considered.

*Reviewer Comment 5: L 24-25 P10. The author conclude that kriging techniques are biased and inaccurate in the tails of the distributions, and prove it with Fig. 4. This is somehow understandable and even quite normal. The kriging techniques are weighted average. Predicting streamflows within a leave-one-out cross-validation, when the lowest or highest streamflow is removed, plus it is perhaps orders of magnitude lower or higher than streamflows from donor sites, it is predictable that the outcome shows upward or downward biases, respectively, in those regimes. I think this thought might be taken into account, or at least pointed out clearly, maybe after those lines or elsewhere in the first sections.*

**Author Response 5:** The reviewer points out that, in many simulations, especially in cross-validation, bias in the tails is an expected result of the calibration framework. I strongly agree and have tried to make this point in lines 5-14 on page 12. I will revise this discussion in the light of this reviewer's and other reviewers' comments. However, a full exploration of the effects of calibration, validation and application of models on tail behavior, as it is equally concerning even outside of the realm of kriging, is conceptually beyond the scope of this paper. Future research is addressing this issue.

**Reviewer Comment 6:** *In case the author adopted unconstrained kriging methods, that is when kriging weights are both positives and negatives, predicting low flows may lead to negative estimates. Would be interesting to know, if any, how many negatives are produced.*

**Author Response 6:** The reviewer is correct that unconstrained kriging methods can produce negative weights. However, because the predictand is a logarithm on streamflow (line 18 of page 5), negative values of streamflow are never produced.

**Reviewer Comment 7:** *By taking the logs of the standardized streamflows, the author implicitly removes zero flows, if any, from the dataset. Is there any catchment with intermittent regime? If so, would be interesting to see how zero flows are treated.*

**Author Response 7:** Zeros were not removed, but were replaced with a value of 0.00003 cubic meters per second, as described in lines 10-13 of page 4. Previous work by Farmer (2015) and Farmer et al. (2014) found this value to have only a limited effect on the comparative results. Additional information on the prevalence of intermittent streams in this data set is available from Farmer et al. (2014) and presented in response to a previous reviewer.

**Reviewer Comment 8:** *L. 33 P9: it is not clear whether or not the author adopted the leave-one-out crossvalidation for the DAR and QPPQ method too. I think so, but I would recommend to rephrase and be a little clearer about the cross-validation procedure used for all the methods reported. Would be even informative to know if any other crossvalidation methods have been used in the past for DAR and QPPQ.*

**Author Response 8:** As the reviewer assumes, leave-one-out cross-validation was used for all methods. I will revise the text to make this more apparent.

**Reviewer Comment 9:** *L. 31-32 P6. I think the parentheses might be removed and extending the text for a few lines might improve the reasoning.*

**Author Response 9:** The parenthesis will be removed. Per recommendations from
another reviewer, this statement will be slightly expanded to explicitly point out the differences between an averaged model and a pooled model.

*Reviewer Comment 10: L. 32 P6, I've noticed the author use to put the punctuation mark before the right hand parenthesis, please correct throughout the manuscript with ")."*

**Author Response 10:** For complete statements or sentences contained within parenthesis, it is correct to place punctuation within the parenthesis. If a fragmentary statement that cannot stand alone as a sentence is made parenthetically within another sentence, the punctuation should be placed outside of the parentheses. I will closely review the usage throughout this manuscript.

*Reviewer Comment 11: L2 P8. "[. . .] developed form", should be "developed from".*

**Author Response 11:** Corrected.

*Reviewer Comment 12: L8 P8. "[. . .] between the 5% and 15% non-exceedance probability" should be perhaps "between the 5% and 15% error"?*

**Author Response 12:** Corrected and clarified per other reviewers' comments.

*Reviewer Comment 13: L30-31 P10. There are two "similarly" adverbs very close one to another. Please, consider to substitute one of them.*

**Author Response 13:** The second usage of 'similarly', appearing on line 31, should have been removed.

*Reviewer Comment 14: Fig. 4 height might be increased. In general Fig. 4 form factor might be changed to improve the readability of the figure itself.*

**Author Response 14:** I will work with the editorial team to ensure readability of this figure.